# An Augmentation-Aware Theory for Self-Supervised Contrastive Learning

**Jingyi Cui** [1]  **Hongwei Wen** [2]  **Yisen Wang** [1,3]

## Abstract

Self-supervised contrastive learning has emerged as a powerful tool in machine learning and computer vision to learn meaningful representations from unlabeled data. Meanwhile, its empirical success has encouraged many theoretical studies to reveal the learning mechanisms. However, in the existing theoretical research, the role of data augmentation is still under-exploited, especially the effects of specific augmentation types. To fill in the blank, we for the first time propose an augmentation-aware error bound for self-supervised contrastive learning, showing that the supervised risk is bounded not only by the unsupervised risk, but also explicitly by a trade-off induced by data augmentation. Then, under a novel semantic label assumption, we discuss how certain augmentation methods affect the error bound. Lastly, we conduct both pixel- and representation-level experiments to verify our proposed theoretical results.

## 1. Introduction

Self-supervised contrastive learning has shown great empirical success in learning representations for computer vision (Chen et al., 2020a;b; He et al., 2020; Chen et al., 2021; Grill et al., 2020; Chen & He, 2021) and multi-modal tasks (Radford et al., 2021; Zhang et al., 2023a). Typically, a contrastive learning model learns to distinguish between similar and dissimilar pairs of data points by drawing near the positive samples (data augmentations of the same instance), while pushing away the negative samples (data augmentations of different instances) (Chen et al., 2020a;b; He et al., 2020; Chen et al., 2021). This process encourages the model

to encode useful features that capture semantic similarities between different instances in the data, leading to improved performance on downstream tasks like classification, detection, or segmentation.

Aside from the empirical success of self-supervised contrastive learning, many theoretical works aim to explain its underlying working mechanisms. The main theoretical framework branches into two categories. The first category directly builds a relationship between the unsupervised contrastive risk and supervised risks via statistical modelings (Arora et al., 2019; Nozawa & Sato, 2021; Ash et al., 2022; Bao et al., 2022; Lei et al., 2023). However, these works assume that the anchor and positive samples are conditionally independent, which contradicts the practical selection procedure of positive samples therefore being unrealistic.

The second category depends on the assumption of augmentation graph and borrow mathematical tools from unsupervised spectral clustering (HaoChen et al., 2021; Zhang et al., 2023a; Zhuo et al., 2023; Wang et al., 2024a;b). However, the augmentation graph assumption is relatively abstract and typically hard to verify. Wang et al. (2021) proposed a similar assumption called augmentation overlap, but it still fails to explain the impacts of specific kinds of data augmentations. Huang et al. (2023) proposed a $(\sigma, \delta)$-measure to mathematically quantify the data augmentation, whereas there still remains a gap between this measure and real-world data augmentation methods.

Aside from these two major categories, there are also other explanatory works of contrastive learning from aspects of feature geometric (Wang & Isola, 2020; Zhang et al., 2023b), information theory (Tian et al., 2020; Wu et al., 2020; Ouyang et al., 2025), independent component analysis (Zimmermann et al., 2021), neighborhood component analysis (Ko et al., 2022), stochastic neighbor embedding (Hu et al., 2023), message passing (Wang et al., 2023), distributionally robust optimization (Wu et al., 2024), etc. Nonetheless, the role of data augmentation is still under-exploited in the existing theoretical frameworks, especially without mathematically analyzing specific data augmentation methods.

Under such background, in this paper, we propose an augmentation-aware theory for self-supervised contrastive learning. Specifically, we derive a novel decomposition of the unsupervised contrastive risk with respect to the number

---

[1]State Key Lab of General Artificial Intelligence, School of Intelligence Science and Technology, Peking University [2]Faculty of Electrical Engineering, Mathematics and Computer Science, University of Twente [3]Institute for Artificial Intelligence, Peking University. Correspondence to: Yisen Wang <yisen.wang@pku.edu.cn>.

*Proceedings of the 42nd International Conference on Machine Learning*, Vancouver, Canada. PMLR 267, 2025. Copyright 2025 by the author(s).

of negative samples sharing the same label with the anchor. Then through investigating the inner risks, we propose our main theorem of the augmentation-aware error bound, which shows that the supervised risk is not only bounded by the unsupervised contrastive risk, but also by the minimum same-class distance and the maximum same-image distance of the augmented dataset. Then under a novel semantic label assumption, we analyze specific types of data augmentation and discuss the existence of a trade-off between the two distance terms with respect to the strength of data augmentation.

Our contributions are summarized as follows.

- We for the first time propose an augmentation-aware error bound for self-supervised contrastive learning, which explicitly includes the quality of data augmentation in the bound without any additional assumptions. The bound shows that data augmentation is equally important to downstream classification as the unsupervised contrastive risk.

- By proposing a novel semantic label assumption, we analyze specific types of data augmentation including random resized crop and color distortion, and show a trade-off with respect to the strength of data augmentation, i.e., the minimum same-class different-image distance decreases while the maximum same-image distance increases as augmentation strength increases.

- We conduct both pixel- and representation-level experiments to verify our theoretical conclusions. We verify the trend of distances with respect to augmentation strength and that the optimal augmentation parameters lead to optimal downstream classification accuracy.

The paper is organized as follows. In Section 2, we propose our theoretical framework, including the mathematical formulations and our proposed augmentation-aware error bound for self-supervised contrastive learning. In Section 3, we compare our bound with existing theoretical studies. Based on the error bound, by proposing the semantic label assumption, we analyze the effects of specific types of data augmentation including random resized crop and random color distortion in Section 4. In Section 5, we conduct numerical experiments to verify our theoretical conclusions. Finally, we conclude our paper in Section 6. Appendices A, B, and C present the related works, proofs of the theorems, and additional experiments, respectively.

## 2. Theoretical Framework

In this section, we first introduce the mathematical formulations in Section 2.1. Then in Section 2.2, we present the main theorem of this paper, i.e. the augmentation-aware error bound. We delay the error analysis of the main theorem

to Section 2.3, where we first propose a novel decomposition of the risk and then investigate the bound of the inner risk. Detailed proofs are shown in Appendix B.

### 2.1. Mathematical Formulations

**Notation.** For original input data, we use the bar notation to denote the original input image, i.e., we let $\bar{X}$ denote the set of all possible data points and denote $P_{\bar{X}}$ as the corresponding distribution. Then we have the input image $\bar{x} \in \mathcal{X} \sim P_{\bar{X}}$. We denote $C \in \mathbb{N}$ as the number of classes, and $c \in [C] \sim \pi_c$ as the class label of $\bar{x}$, where $[C] := \{1, \ldots, C\}$ and $\pi_c := P(y = c)$ denotes the marginal probability distribution of Class $c$. Denote $\boldsymbol{\pi} = \{\pi_c\}_{c=1}^C$. Moreover, we denote $\rho_c := P(\cdot|y = c)$ as the posterior probability distribution of Class $c$. For the augmented data, we let $\mathcal{A}$ denote the set of all possible data augmentations, and denote $P_A$ as the corresponding distribution. We denote $x := a(\bar{x})$ as the augmented image, where $a \in \mathcal{A} \sim P_A$.

**Unsupervised contrastive learning.** In unsupervised contrastive learning, we select the different data augmentations of the same input image as the positive samples, and select the data augmentations of different input images as negative samples. The data generation process is described as follows: (i) draw positive/negative classes: $c$, $\{c_k^-\}_{k=1}^K \sim \boldsymbol{\pi}^{K+1}$; (ii) draw an original sample for the anchor and positives $\bar{x} \sim \rho_c$; (iii) draw original samples for the negatives $\bar{x}_k^- \sim \rho_{c_k}, k = 1, \ldots, K$; (iv) draw data augmentations $a, a', \{a_k\}_{k=1}^K \sim \mathcal{A}^{K+1}$. Then the anchor is $x = a(\bar{x})$, the positive sample is $x' = a'(\bar{x})$, and the negative samples are $x_k = a_k(\bar{x}_k), k = 1, \ldots, K$.

Compared with the CURL frameworks (Arora et al., 2019), our formulation accords better with the practical applications of unsupervised contrastive learning, where the anchor and positive samples are drawn from the augmentations of the same input image, instead of the same latent class samples. That means, we no longer need the conditional independence assumption. Moreover, we formulate the data augmentations $a, a', \{a_k\}_{k=1}^K$ explicitly, making it possible to analyze the impacts of data augmentation to the error bounds.

In this paper, we investigate the widely used InfoNCE loss function (Chen et al., 2020a; He et al., 2020; Arora et al., 2019), i.e., for $f : \mathcal{X} \to \mathbb{R}^d$, the loss function is

$$\mathcal{L}^{\mathrm{un}}(x, x', \{x_k\}_{k=1}^K; f)$$
$$:= -\log\left(\frac{e^{f(x)^\top f(x')}}{e^{f(x)^\top f(x')} + \sum_{k=1}^K e^{f(x)^\top f(x_k)}}\right), \quad (1)$$

which is mathematically equivalent to the logistic form

$$\mathcal{L}^{\mathrm{un}}(x, x', \{x_k\}_{k=1}^K; f)$$

$$:= \log\Big(1 + \sum_{k=1}^{K} \exp\big(-f(x)^{\top}[f(x') - f(x_k)]\big)\Big). \quad (2)$$

Under the above-mentioned data generation process, we have the unsupervised contrastive risk as

$$\mathcal{R}^{\mathrm{un}}(f) := \mathbb{E}_{c,\{c_k\}_{k\in[K]}} \mathbb{E}_{\bar{x}\sim\rho_c, \bar{x}_k\sim\rho_{c_k}} \mathbb{E}_{a,a',\{a_k\}_{k\in[K]}}$$
$$\cdot \mathcal{L}^{\mathrm{un}}(a(\bar{x}), a'(\bar{x}), \{a_k(\bar{x}_k)\}_{k=1}^{K}; f). \quad (3)$$

For the empirical forms, define the datasets

$$S := \{(x_j, x'_j, x_{j1}, \ldots, x_{jK})\}_{j\in[n]}.$$

Then the empirical unsupervised risk is denoted as

$$\widehat{\mathcal{R}}^{\mathrm{un}}(f) := \frac{1}{n} \sum_{j\in[n]} \mathcal{L}^{\mathrm{un}}(x_j, x'_j, \{x^-_{jk}\}_{k=1}^{K}; f) \quad (4)$$

**Downstream supervised classification.** To evaluate the representations learned by unsupervised contrastive learning, we adopt the linear-probing setting, i.e. given the learned representation $f : \mathcal{X} \to \mathbb{R}^d$, we train a linear classifier $g = \boldsymbol{W}f : \mathbb{R}^d \to \mathbb{R}^C$ on top of $f$ with $\boldsymbol{W} \in \mathbb{R}^{C\times d}$. Note that in the downstream classification task, usually the goal is to classify original images instead of the augmented ones, because when directly using a pre-trained model, it is hard to know the exact augmentation methods in training and replicate them on the downstream datasets.

Following previous theoretical works (Arora et al., 2019; Nozawa & Sato, 2021; Ash et al., 2022; Bao et al., 2022), we use the mean classifier for evaluation. Specifically, the mean classifier is defined as $g = \boldsymbol{W}f$, where $\boldsymbol{W} := [\mu_1, \ldots, \mu_C]^{\top}$, $\mu_c := \mathbb{E}_{\bar{x}\in\rho_c} f(\bar{x})$, $c \in [C]$.

We adopt the softmax cross entropy loss for the mean classifier $g$, defined by

$$\mathcal{L}^{\mathrm{sup}}(\bar{x}, c; f) = -\log\Big(\frac{e^{g(\bar{x})_c}}{\sum_{i=1}^{K} e^{g(\bar{x})_i}}\Big), \quad (5)$$

which is mathematically equivalent to the logistic form

$$\mathcal{L}^{\mathrm{sup}}(\bar{x}, c; f) = \log\Big(1 + \sum_{c_k\neq c} \exp\big(-f(\bar{x})^{\top}(\mu_c - \mu_{c_k})\big)\Big).$$

Under the data generation process, we have the supervised risk as

$$\mathcal{R}^{\mathrm{sup}}(f) := \mathbb{E}_{c\sim\boldsymbol{\pi}} \mathbb{E}_{\bar{x}\sim\rho_c} \mathcal{L}^{\mathrm{sup}}(\bar{x}, c; f). \quad (6)$$

### 2.2. Augmentation-Aware Error Bound

In Theorem 2.1, we present the upper bound of the supervised risk of the downstream mean classifier (the lower bound of the unsupervised contrastive risk), where two terms regarding with data augmentations are shown explicitly in the bound.

**Theorem 2.1** (Error Bound). *Let $\mathcal{R}^{\mathrm{sup}}(f)$ be the supervised risk of the mean classifier, and $\mathcal{R}^{\mathrm{un}}(f)$ be the unsupervised risk of contrastive loss. Denote $\tau_K = \mathrm{P}(\mathrm{Col}(c, \{c_k\}_{k=1}^{K}) \neq 0)$ as the class collision probability, where $\mathrm{Col}(c, \{c_k\}_{k=1}^{K}) = \sum_{k=1}^{K} \mathbf{1}[c = c_k]$. Then we have the following upper bound of the supervised risk.*

$$\mathcal{R}^{\mathrm{sup}} \leq \frac{1}{1-\tau_K}[\mathcal{R}^{\mathrm{un}} - \tau_K \mathbb{E}_{c,\{c_k\}_{k=1}^{K}} \log(\mathrm{Col} + 1)$$
$$+ \mathbb{E}_c \mathbb{E}_{\bar{x},\bar{x}'\sim\rho_c} \mathbb{E}_a \min_{a'} \|f(a(\bar{x})) - f(a'(\bar{x}'))\|$$
$$+ 5\mathbb{E}_c \mathbb{E}_{\bar{x}'\sim\rho_c} \max_{a,a'} \|f(a(\bar{x}')) - f(a'(\bar{x}'))\|]. \quad (7)$$

Theorem 2.1 shows that the downstream supervised risk is upper bounded by both the unsupervised contrastive risk, CURL's class collision term, and two distance terms. The first term represents the minimum distance between two augmented same-class (different) images. Intuitively, this term measures how well the same-class images are connected. The better the same-class connection, the smaller this term is. The second distance term represents the maximum distance between the two augmentations of the same images. Intuitively, this term could be understood as the range or variance of data augmentation.

Typically, as the augmentation strength increases, an image is more likely to be connected with other images under data augmentation, and the first distance term is smaller. On the other hand, stronger augmentation has larger range and variance. As a result, there is a trade-off between the two distance terms w.r.t. augmentation strength. This point will be further discussed in greater detail in Section 4 and verified in Section 5.

In addition, it is worthwhile mentioning that we can further improve the coefficient of the second distance term to 1 with a mild assumption that the original input image has a centered representation among all of its augmentations. This assumption has been made implicitly by previous works, e.g. Nozawa & Sato (2021); Zimmermann et al. (2021).

**Assumption 2.2** (Centered Representation). *For $\bar{x} \in \mathcal{X}$ and $f : \mathcal{X} \to \mathbb{R}^d$, we assume that $\mathbb{E}_{a\sim\mathrm{P}_A} f(a(\bar{x})) = f(\bar{x})$.*

**Theorem 2.3** (Error Bound (Improved)). *Under Assumption 2.2, we have*

$$\mathcal{R}^{\mathrm{sup}} \leq \frac{1}{1-\tau_K}[\mathcal{R}^{\mathrm{un}} - \tau_K \mathbb{E}_{c,\{c_k\}_{k=1}^{K}} \log(\mathrm{Col} + 1)$$
$$+ \mathbb{E}_c \mathbb{E}_{\bar{x},\bar{x}'\sim\rho_c} \mathbb{E}_a \min_{a'} \|f(a(\bar{x})) - f(a'(\bar{x}'))\|$$
$$+ \mathbb{E}_c \mathbb{E}_{\bar{x}'\sim\rho_c} \max_{a,a'} \|f(a(\bar{x}')) - f(a'(\bar{x}'))\|]. \quad (8)$$

To provide the generalization bound, we introduce the empirical Rademacher complexity as follows,

$$\mathrm{Rad}_S(\mathcal{F}) := \mathbb{E}_{\epsilon\sim\{\pm1\}^{3nKd}} \mathbb{E}\Big(\sup_{f\in\mathcal{F}} \sum_{j\in[n]} \sum_{k\in[K]} \sum_{t\in[d]}$$

$$\left(\epsilon_{j,k,t,1} f_t(x_j) + \epsilon_{j,k,t,2} f_t(x'_j) + \epsilon_{j,k,t,3} f_t(x_{jk})\right)$$

**Theorem 2.4** (Generalization Bound). *Assume that* $\|f(x)\|_2 \leq R$ *for any* $f \in \mathcal{F}$ *and* $x \in \mathcal{X}$ *and the unsupervised loss* $\mathcal{L}^{\mathrm{un}}$ *is bounded by* $B$. *Then for any* $\delta \in (0,1)$ *and any* $f \in \mathcal{F}$, *we have*

$$
\mathcal{R}^{\mathrm{sup}}(f) \leq \frac{1}{1 - \tau_K}[\widehat{\mathcal{R}}^{\mathrm{un}}(f) + \frac{12R\mathrm{Rad}_S(\mathcal{F})}{n}
$$
$$
+ 3B\sqrt{\frac{\log(2\delta)}{2n}} - \tau_K \mathbb{E}_{c,\{c_k\}_{k=1}^K} \log(\mathrm{Col} + 1)
$$
$$
+ \mathbb{E}_c \mathbb{E}_{\bar{x},\bar{x}' \sim \rho_c} \mathbb{E}_a \min_{a'} \|f(a(\bar{x})) - f(a'(\bar{x}'))\|
$$
$$
+ 5\mathbb{E}_c \mathbb{E}_{\bar{x}' \sim \rho_c} \max_{a,a'} \|f(a(\bar{x}')) - f(a'(\bar{x}'))\|].
$$

*with probability at least* $1 - \delta$.

We show the generalization bound in Theorem 2.4, where the population downstream supervised risk is bounded by the empirical unsupervised contrastive risk. Note that the proof follows from Lei et al. (2023), where the dependence of $K$ is removed compared with that in Arora et al. (2019).

### 2.3. Error Analysis

In this part, we present the key theorems in proving Theorem 2.1, the main theorem of this paper. Specifically, in Theorem 2.5, we derive a novel decomposition of the unsupervised contrastive risk w.r.t. the number of negative samples sharing the same label with the anchor. Note that this decomposition is non-trivial, as it only works for contrastive losses that treat negative samples equally.

**Theorem 2.5** (Error Decomposition of $\mathcal{R}^{\mathrm{un}}$). *We have*

$$
\mathcal{R}^{\mathrm{un}}(f)
$$
$$
= \mathbb{E}_c \mathbb{E}_{\bar{x} \sim \rho_c} \mathbb{E}_a \sum_{i_1 \neq c} \cdots \sum_{i_K \neq c} p_K(i_1, \ldots, i_K) r_K(i_1, \ldots, i_K)
$$
$$
\cdots
$$
$$
+ \mathbb{E}_c \mathbb{E}_{\bar{x} \sim \rho_c} \mathbb{E}_a \sum_{i_j \neq c} p_1(i_j) r_1(i_j) + \mathbb{E}_c \mathbb{E}_{\bar{x} \sim \rho_c} \mathbb{E}_a p_0 r_0, \quad (9)
$$

*where for* $k = 0, \ldots, K$ *and* $i_1, \ldots, i_k$, *we denote*

$$
r_k(i_1, \ldots, i_k) := \mathbb{E}_{\bar{x}_1 \sim \rho_{i_1}} \cdots \mathbb{E}_{\bar{x}_k \sim \rho_{i_k}} \mathbb{E}_{\bar{x}_{k+1}, \ldots, \bar{x}_K \sim \rho_c}
$$
$$
\cdot \mathbb{E}_{a', \{a_k\}_{k \in [K]}} \mathcal{L}(a(\bar{x}), a'(\bar{x}), a_k(\bar{x}_k); f), \quad (10)
$$

*and*

$$
p_k(i_1, \ldots, i_k) := \mathrm{P}(\exists \{j_1, \ldots, j_K\},
$$
$$
\text{such that } c_{j_1} = i_1, \ldots, c_{j_k} = i_k,
$$
$$
\text{and } c_{j_{k+1}} = \cdots = c_{j_K} = c), \quad (11)
$$

*where* $\{j_1, \ldots, j_K\}$ *is a rearrangement of* $[K]$.

To derive the relationship between the unsupervised contrastive risk $\mathcal{R}^{\mathrm{un}}(f)$ and the downstream classification risk $\mathcal{R}^{\mathrm{sup}}(f)$, we construct an intermediate supervised risk $\bar{\mathcal{R}}^{\mathrm{sup}}(f) := \mathbb{E}_{c,\{c_k\}_{k=1}^K} \mathbb{E}_{\bar{x} \sim \rho_c} \log\left(1 + \sum_{k=1}^K \exp\left(-f(\bar{x})^\top(\mu_c - \mu_{c_k})\right)\right)$, which has a similar decomposition with $\mathcal{R}^{\mathrm{un}}(f)$ shown in Corollary B.1.

Next, in Theorem 2.6, we investigate each inner risks $r_k$, and build a relationship between $r_k$ and $r_k^{\mathrm{sup}}$. We show that for $k = 0, \ldots, K$, $r_k$ is upper bounded by $r_k^{\mathrm{sup}}$ minus a series of distance terms, including the distances between same-class different-image augmentations and that between same-image data augmentations.

**Theorem 2.6** (Bound of Inner Risk). *Let* $\bar{x}$ *belong to Class* $c$. *For* $k = 0, \ldots, K$, *given* $i_1, \ldots, i_k \neq c$, *we have*

$$
r_k(i_1, \ldots, i_k)
$$
$$
\geq r_k^{\mathrm{sup}}(i_1, \ldots, i_k) - [2\|f(a(\bar{x})) - f(\bar{x})\|
$$
$$
+ 2\mathbb{E}_{\bar{x}' \sim \rho_c} \max_{a',a''} \|f(a''(\bar{x}')) - f(a'(\bar{x}'))\|
$$
$$
+ \mathbb{E}_{\bar{x}_m \sim \rho_{i_m}} \max_{a',a''} \|f(a''(\bar{x}_m)) - f(a'(\bar{x}_m))\|
$$
$$
+ \mathbb{E}_{\bar{x}' \sim \rho_c} \mathbb{E}_{a'} \min_{a''} \|f(a'(\bar{x})) - f(a''(\bar{x}'))\|], \quad (12)
$$

*where* $r_k^{\mathrm{sup}}(i_1, \ldots, i_k) := \log\left(1 + \exp\left(-\sum_{m=1}^k f(\bar{x})^\top(\mu_c - \mu_{i_m})\right)\right)$ *and* $\mu_i = \mathbb{E}_{\bar{x}' \sim \rho_i} f(\bar{x}')$, $i \in \{c, i_1, \ldots, i_k\}$.

Then combining Theorem 2.5, Corollary B.1, and Theorem 2.6, we reach Theorem 2.7 showing the relationship between $\mathcal{R}^{\mathrm{un}}$ and $\bar{\mathcal{R}}^{\mathrm{sup}}$, i.e. $\bar{\mathcal{R}}^{\mathrm{sup}}$ is upper bounded by $\mathcal{R}^{\mathrm{un}}$ and two distance terms induced by data augmentation.

**Theorem 2.7** (Error Bound of $\bar{\mathcal{R}}^{\mathrm{sup}}$). *We have the following upper bound of the supervised risk.*

$$
\bar{\mathcal{R}}^{\mathrm{sup}} \leq \mathcal{R}^{\mathrm{un}} + \mathbb{E}_c \mathbb{E}_{\bar{x},\bar{x}' \sim \rho_c} \mathbb{E}_a \min_{a'} \|f(a(\bar{x})) - f(a'(\bar{x}'))\|
$$
$$
+ 5\mathbb{E}_c \mathbb{E}_{\bar{x}' \sim \rho_c} \max_{a,a'} \|f(a(\bar{x}')) - f(a'(\bar{x}'))\|. \quad (13)
$$

Finally, to close the gap between $\bar{\mathcal{R}}^{\mathrm{sup}}$ and $\mathcal{R}^{\mathrm{sup}}$, we adopt the CURL bound directly. Theorem 2.7 and Lemma 2.8 together finish the proof of Theorem 2.1.

**Lemma 2.8** (CURL bound (Arora et al., 2019; Nozawa & Sato, 2021)). *Denote* $\tau_K = \mathrm{P}(\mathrm{Col}(c, \{c_k\}_{k=1}^K) \neq 0)$ *as the class collision probability, where* $\mathrm{Col}(c, \{c_k\}_{k=1}^K) = \sum_{k=1}^K \mathbf{1}[c = c_k]$. *Then we have*

$$
\bar{\mathcal{R}}^{\mathrm{sup}}(f) = (1 - \tau_K)\mathcal{R}^{\mathrm{sup}}(f)
$$
$$
+ \tau_K \mathbb{E}_{c,\{c_k\}_{k=1}^K} \log(\mathrm{Col} + 1). \quad (14)
$$

## 3. Discussions on the Error Bound

We first compare our bound with previous studies on building relationships between the unsupervised contrastive risk

and downstream supervised risk. Most of these previous bounds are under the CURL framework (Arora et al., 2019; Ash et al., 2022; Bao et al., 2022), which assumes that the positive samples are conditionally independently drawn from a certain class, whereas our framework explicitly formulates the probability distribution of data augmentation and assumes that the positive samples are drawn as different augmentations of the same input image. That is, we are based on a more realistic data generation process that accords with the empirical applications of contrastive learning.

Perhaps more related to our bound, Nozawa & Sato (2021) derived a lower bound of InfoNCE loss based on the same data generation process as ours. However, they treated the gap term $d(f)$ induced by data augmentation as an almost constant, whereas in this paper, we further dissect the augmentation-relevant terms into two distance terms based on our proposed error decomposition, resulting in a trade-off between the minimum same-class distance and the maximum same-image distance. Our result suggests that to reach a better downstream classification risk, data augmentation is equally important as the unsupervised contrastive risk.

Also note that because the focus of our bound is on the role of data augmentation rather than that of negative samples, we directly adopt CURL's class collision term for simplicity. Nonetheless, our bound is in fact compatible with previous bounds under the CURL framework, e.g. Arora et al. (2019); Nozawa & Sato (2021); Ash et al. (2022); Bao et al. (2022), with replacing the class collision term with their corresponding forms. As the two distance terms in (7) are independent of the number of negative samples $K$, we could reach the same conclusions on the role of $K$ as in the corresponding previous works.

We also compare our bound with works that explain contrastive learning from the perspective of data augmentation. Wu et al. (2020) discussed the importance of data augmentation from the perspective of permutation invariance, but the impact of data augmentation was not reflected in the error bound. HaoChen et al. (2021) proposed a concept called augmentation graph where the vertices are all the augmented data points and the edge weights is the probabilities that two augmentations are generated from the same input image. Their derived downstream error bound relies on the eigenvalues of the adjacency matrix of the augmentation graph. Similarly, Wang et al. (2021) built their theory on the assumption of augmentation overlap, meaning that two images have the same augmented view under some data augmentation. However, it is unlikely in practice that two different real images have exactly the same augmented views, especially as we usually use only two views in training instead of using multiple ones. By contrast, in this paper, we treat the connectivity between augmented views in a rather "soft" way. Specifically, our bound does not require the augmen-

tations to be exactly the same, nor do we need the perfect alignment assumption adopted in Wang et al. (2021). Instead, the distances between different augmentations are shown explicitly in our bound without any additional assumptions. Later on, Huang et al. (2023) defined a kind of $(\sigma, \delta)$-measure to mathematically quantify the data augmentation, and provided an upper bound of the downstream classification error rate. Whereas their bound relies on the maximum (augmentation) distance of same-class images, our bound relies on a trade-off between two distance terms, enabling us to further discuss real-world data augmentation methods.

## 4. Impacts of Data Augmentations

In this section, we try to explain the trade-off between the two distance terms in Theorems 2.1 and 2.3 from the perspective of pixel-level discussions of the training images. As suggested by Chen et al. (2020a), the combination of random cropping and random color distortion contributes to successful downstream classification. Therefore, in this part, we discuss the impacts of random cropping and random color distortion respectively.

First of all, under the Lipschitz continuous assumption, we transform our main theory into an error bound w.r.t. the pixel-level distances, so that we are able to discuss the impact of specific data augmentation.

**Assumption 4.1** (Lipschitz Continuity). For $x, x' \in \mathcal{X}$, there exists a constant $c_L \geq 0$, such that

$$\|f(x) - f(x')\| \leq c_L \|x - x'\|. \tag{15}$$

**Theorem 4.2** (Error Bound with Pixel-level Distances). *Under Assumptions 2.2 and 4.1, we have*

$$\begin{aligned} \mathcal{R}^{\mathrm{sup}} \leq \frac{1}{1 - \tau_K} [ & \mathcal{R}^{\mathrm{un}} - \tau_K \mathbb{E}_{c, \{c_k\}_{k=1}^K} \log(\mathrm{Col} + 1) \\ & + c_L \mathbb{E}_c \mathbb{E}_{\bar{x}, \bar{x}' \sim \rho_c} \mathbb{E}_a \min_{a'} \|a(\bar{x}) - a'(\bar{x}')\| \\ & + c_L \mathbb{E}_c \mathbb{E}_{\bar{x}' \sim \rho_c} \max_{a, a'} \|a(\bar{x}') - a'(\bar{x}')\|]. \tag{16} \end{aligned}$$

Inspired by Koenderink (1984), we introduce the pixel-level formulation of an $d \times d$ 3-channel original input image. Specifically, for any integers $j, \ell \in [d]$, define

$$I_{j,\ell} = [\frac{j-1}{d}, \frac{j}{d}) \times [\frac{\ell-1}{d}, \frac{\ell}{d}) \tag{17}$$

representing a square with side length $1/d$. Then in each channel, a $d \times d$ image $\bar{x} = (\bar{x}_{j,\ell}^{(i)})_{j,\ell \in [d], i=1,2,3}$ with $d^2$ pixels can be expressed as a function

$$\xi^{(i)} : \mathbb{R}^2 \to [0, \infty), \tag{18}$$

where the value of the $(j,\ell)$-th pixel in the $i$-th channel is given by

$$\bar{\xi}_{j,\ell}^{(i)} = d^2 \int_{I_{j,\ell}} \xi^{(i)}(u,v)\,du\,dv, \quad j,\ell \in \{1,\ldots,d\}, \quad (19)$$

representing the average intensity of $\xi^{(i)}$ on $I_{j,\ell}$.

### 4.1. Semantic Label Assumption

We introduce the generation process of pixels based on a semantic label assumption. We underline that our proposed pixel-level generation process aims just to provide a possible explanation for the trade-off in the two distances terms of Theorem 4.2, rather than generating images.

Specifically, we assume that an image can be decomposed into several *semantic areas* with their corresponding *semantic labels*, where the semantic areas/labels are dependent on the class label. That is, for an original image $\bar{x}$, aside from a class label $y$ for the entire image, each pixel $\bar{h}_{j,\ell}$ also has a unique semantic label $s$ that relates to $y$.

For example, an image of Class *automobile* usually has semantic features *windshield*, *headlights*, *wheels*, etc., and an image of Class *truck* usually has se-mantic features *truck cab*, *cargo box*, *wheels*, etc. As illustrated in Figure 1, we mark the disjoint semantic areas with distinguished color boxes, and we mark the shared se-mantic *wheels* with the same color.

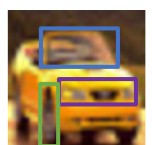 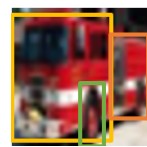

(a) Automobile. (b) Truck.

*Figure 1.* Illustration of semantic label assumption. (a) An *automobile* image with semantic labels *windshield* (blue), *headlights* (pur-ple), and *wheels* (green); (b) an *truck* image with semantic labels *truck cab* (yellow excluding green), *cargo box* (orange), and *wheels* (green).

Mathematically, assume there are $T > 0$ possible semantic labels $\{s_1,\ldots,s_T\} \in \{0,1\}^T$. (Without loss of generality, we assume $s_T$ represents the background). We use $s_t = 1$ to describe that the semantic $s_t$ exists in an image of Class $y$ and $s_t = 0$ otherwise. For $t \in [T]$ and $y \in [C]$, we denote $q_y := \mathrm{P}(s_t = 1|y)$ as the probability that a semantic label $s_t$ is related to Class label $y$. Moreover, for $i = 1, 2, 3$, denote $\eta_{s_t}^{(i)} = \mathrm{P}(\bar{\xi}^{(i)}|s_t)$ as the probability of a pixel with semantic label $s_t$ to have value $\bar{\xi}^{(i)}$ in the $i$-th channel.

Then the pixel-level generation process is shown as follows: (i) Draw a class label $y \in [C] \sim \boldsymbol{\pi}$; (ii) draw semantic labels $s_t \sim q_y$; (iii) generate a disjoint random partition $(J_m)_{m=1}^{\sum_{t=1}^{T} s_t}$; (iv) For each pixel in $J_m$ with semntic label $s_t$, draw a value in the $i$-th channel following $\eta_{s_t}^{(i)}$.

### 4.2. Impacts of Random Crop

Based on the pixel-level formulation, given the scale param-eter $\delta \in (0,1]$, we formulate the random resized crop of an image as $\xi \circ A_{\mathrm{crop}}$

$$\xi \circ a_{\mathrm{crop}}(u,v) = \xi(a_{\mathrm{crop}}(u,v)) = \xi(\theta u - \tau, \theta v - \tau'),$$

where $\theta \sim \mathrm{Unif}(0,\delta]$, $\tau, \tau' \sim \mathrm{Unif}[0,1]$ are i.i.d. random variables.

Denote $\|\cdot\|_F$ as the Frobenius norm. For the minimum dis-tance between same-class different-image augmentations, if $a(\bar{x})$ contains only same-semantic label pixels (with seman-tic label $s$), then we have

$$\mathbb{E}_{a(\bar{x})} \min_{a'} \|a(\bar{x}) - a'(\bar{x}')\|_F$$

$$= \mathbb{E}_{\bar{\xi}} \min_{a'} \Big[ \sum_{j,\ell \in [d], i \in [3]} \big(\bar{\xi}_{j,\ell}^{(i)} - \bar{\xi}'_{j,\ell}^{(i)}\big)^2 \Big]^{1/2}$$

$$\leq \Big[ \sum_{j,\ell \in [d], i \in [3]} \mathbb{E}_{\bar{\xi}_{j,\ell}^{(i)} \sim \eta_s^{(i)}} \big(\bar{\xi}_{j,\ell}^{(i)} - \mu_s^{(i)}\big)^2 \Big]^{1/2}$$

$$+ \Big[ \sum_{j,\ell \in [d], i \in [3]} \mathbb{E}_{\bar{\xi}'_{j,\ell}^{(i)} \sim \eta_s^{(i)}} \big(\bar{\xi}'_{j,\ell}^{(i)} - \mu_s^{(i)}\big)^2 \Big]^{1/2}$$

$$= 2\Big[ d^2 \cdot \sum_{i \in [3]} \sigma_s^{(i)2} \Big]^{1/2} := 2\sigma, \quad (20)$$

where $\mu_s^{(i)}$ and $\sigma_s^{(i)2}$ denote the mean and variance pixel value of semantic class $s$ in the $i$-th channel, and the second inequality holds because there exists a small enough $\theta$ such that its induced resized-cropped $a'(x')$ has all pixels with semantic label $s$.

On the other hand, if the pixels in $a(\bar{x})$ has more than one semantic labels, then we have

$$\mathbb{E}_{a(\bar{x})} \min_{a'} \|a(\bar{x}) - a'(\bar{x}')\|_F$$

$$= \mathbb{E}_{\bar{\xi}} \min_{a'} \Big[ \sum_{s} \sum_{s(\bar{\xi}_{j,\ell})=s, i \in [3]} \big(\bar{\xi}_{j,\ell}^{(i)} - a'(\bar{x}')_{j,\ell}^{(i)}\big)^2 \Big]^{1/2}$$

$$\leq \Big[ \sum_{j,\ell \in [d], i \in [3]} \mathbb{E}_{\bar{\xi}_{j,\ell}^{(i)} \sim \eta_s^{(i)}} \big(\bar{\xi}_{j,\ell}^{(i)} - \mu_s^{(i)}\big)^2 \Big]^{1/2}$$

$$+ \Big[ \sum_{j,\ell \in [d], i \in [3]} \mathbb{E}_{\bar{\xi}'_{j,\ell}^{(i)} \sim \eta_{s_{\max}}^{(i)}} \big(\bar{\xi}'_{j,\ell}^{(i)} - \mu_{s_{\max}}^{(i)}\big)^2 \Big]^{1/2}$$

$$+ \Big[ \sum_{s(\bar{\xi}_{j,\ell}) \neq s_{\max}, i \in [3]} \big(\mu_s^{(i)} - \mu_{s_{\max}}^{(i)}\big)^2 \Big]^{1/2}$$

$$:= 2\sigma + \Big[ \sum_{j,\ell \in [d]} \mathbf{1}[s(\bar{\xi}_{j,\ell}) \neq s_{\max}] \cdot \Delta\mu^2 \Big]^{1/2}, \quad (21)$$

where $s_{\max}$ denotes the majority semantic label that most pixels in $a(\bar{x})$ have. Compared with the single-semantic

case (20), there is an additional bias term in (21) resulting in a larger overall distance. Moreover, the larger the crop size, the larger the probability that the cropping area intersects with the partition boundary, leading to a higher probability that $a(\bar{x})$ has more semantic labels and larger $\sum_{j,\ell\in[d]} \mathbf{1}[s(\bar{\xi}_{j,\ell}) \neq s_{\max}]$. Consequently, a larger crop size (or larger scale parameter $\delta$) results in larger value of $\mathbb{E}_c \mathbb{E}_{\bar{x},\bar{x}'\sim\rho_c} \mathbb{E}_a \min_{a'} \|a(\bar{x}) - a'(\bar{x}')\|$.

Likewise, for the maximum distance between same-image different augmentations, with a smaller crop size, it is more likely for $a(\bar{x})$ to have different pixel-level semantic labels with the least alike augmentation $a'(\bar{x})$, and therefore having larger bias term and the overall distance $\mathbb{E}_c \mathbb{E}_{\bar{x}'\sim\rho_c} \max_{a,a'} \|a(\bar{x}') - a'(\bar{x}')\|$.

### 4.3. Impacts of Color Distortion

The random color distortion used for data augmentation usually contains adjustments of brightness, contrast, saturation, and hue. For simplicity, we use brightness manipulation of each channel as an example to represent the random augmentations w.r.t. colors. Specifically, based on the pixel-level formulation, given the brightness adjustment parameter $b > 0$, we formulate the random brightness distortion of an image as $f \circ a_{\mathrm{color}}$

$$\xi^{(i)} \circ a_{\mathrm{color}}(u,v) = \lambda^{(i)} \cdot \xi^{(i)}(u,v), \qquad (22)$$

where $\lambda^{(i)} \in \mathrm{Unif}(0,b)$, $i \in [3]$.

With combining $a_{\mathrm{color}}$ and $a_{\mathrm{crop}}$, if $a(\bar{x})$ contains only same-semantic label pixels (with semantic label $s$), then we have

$$\mathbb{E}_{a(\bar{x})} \min_{a'} \|a(\bar{x}) - a'(\bar{x}')\|_F$$
$$\leq \Big[ \sum_{j,\ell\in[d],i\in[3]} \mathbb{E}_{\bar{\xi}^{(i)}_{j,\ell}\sim\eta^{(i)}_s} \big(\bar{\xi}^{(i)}_{j,\ell} - \mu^{(i)}_s\big)^2 \Big]^{1/2}$$
$$+ \Big[ \sum_{j,\ell\in[d],i\in[3]} \mathbb{E}_{\bar{\xi}'^{(i)}_{j,\ell}\sim\eta^{(i)}_s} \big(\lambda^{(i)} \cdot \bar{\xi}'^{(i)}_{j,\ell} - \mu^{(i)}_s\big)^2 \Big]^{1/2}$$
$$= \Big[ d^2 \cdot \sum_{i\in[3]} \sigma^{(i)2}_s \Big]^{1/2} := \sigma, \qquad (23)$$

where the equality holds by taking $\lambda^{(i)} = \mu^{(i)}_s/\bar{\xi}^{(i)}_{j,\ell}$. Compared with (20), the combination of color distortion further reduces the minimum same-class different-image distance by half. Similarly, when the pixels in $a(\bar{x})$ have more than one semantic labels, we have

$$\mathbb{E}_{a(\bar{x})} \min_{a'} \|a(\bar{x}) - a'(\bar{x}')\|_F$$
$$\leq \Big[ \sum_{j,\ell\in[d],i\in[3]} \mathbb{E}_{\bar{\xi}^{(i)}_{j,\ell}\sim\eta^{(i)}_s} \big(\bar{\xi}^{(i)}_{j,\ell} - \mu^{(i)}_s\big)^2 \Big]^{1/2}$$
$$+ \Big[ \sum_{j,\ell\in[d],i\in[3]} \mathbb{E}_{\bar{\xi}^{(i)}_{j,\ell}\sim\eta^{(i)}_{s_{\max}}} \big(\lambda^{(i)}\bar{\xi}'^{(i)}_{j,\ell} - \mu^{(i)}_{s_{\max}}\big)^2 \Big]^{1/2}$$

$$+ \Big[ \sum_{s(\bar{\xi}_{j,\ell})\neq s_{\max},i\in[3]} (\mu^{(i)}_s - \mu^{(i)}_{s_{\max}})^2 \Big]^{1/2}$$
$$:= \sigma + \Big[ \sum_{j,\ell\in[d]} \mathbf{1}[s(\bar{\xi}_{j,\ell}) \neq s_{\max}] \cdot \Delta\mu^2 \Big]^{1/2}, \qquad (24)$$

where the last equation holds by taking $\lambda^{(i)} = \mu^{(i)}_s/\bar{\xi}^{(i)}_{j,\ell}$. Compared with the uniform-semantic label case, when the pixels have multiple semantic labels, the color manipulation has a less significant effect because it can only reduce the distance terms with regard to $s_{\max}$ and still fails to deal with the bias term. This also to some extent explains that without cropping, color distortion alone does not lead to good downstream performance. (As shown in Figure 5 of Chen et al. (2020a), the linear probing accuracy of color+crop is 56.3, whereas that of color alone is merely 18.8.)

On the other hand, for the maximum same-image distance, by (21), color distortion enhances the bias term, because if $b > 1$, we have

$$\max_{a,a'} \|a(\bar{x}) - a'(\bar{x})\|_F$$
$$= \max_{\lambda^{(i)}\in(0,b)} \Big[ \sum_{j,\ell\in[d],i\in[3]} \big(\bar{\xi}^{(i)}_{j,\ell} - \lambda^{(i)}\bar{\xi}'^{(i)}_{j,\ell}\big)^2 \Big]^{1/2}$$
$$\geq \Big[ \sum_{j,\ell\in[d],i\in[3]} \big(\bar{\xi}^{(i)}_{j,\ell} - \bar{\xi}'^{(i)}_{j,\ell}\big)^2 \Big]^{1/2}. \qquad (25)$$

## 5. Verification Experiments

In this section, we aim to verify our theoretical conclusions through empirical experiments. Specifically, in Section 5.1, we verify the trade-off between the two distance terms induced by data augmentation in Theorem 2.1. Then in Section 5.2, we show that the parameters of data augmentation minimizing the two distance terms coincides with the optimal parameters for downstream accuracy, which in turn verifies the effectiveness of Theorem 2.3.

**Experimental Setup.** We conduct numerical comparisons on the CIFAR-100 and TinyImagenet benchmark datasets. For conciseness of presentation, we delay the figures regarding TinyImagenet to Appendix C. We follow the experimental settings of SimCLR (Chen et al., 2020a). Specifically, we use ResNet-18 as the backbone and a 2-layer MLP as the projection head. We set the batch size as 1024 and use 1000 epochs for training representations. We use the SGD optimizer with the learning rate 0.5 decayed at the 700-th, 800-th, and 900-th epochs with a weight decay 0.1. We run all experiments on an NVIDIA GeForce RTX 3090 24GB GPU.

The data augmentations we use are random resized crop, random horizontal flip, random color jitter, and random grayscale. The default crop size $\in [0.2, 1.0]$, the default

flip probability is 0.5, the default color probability is 0.8, and the default gray probability is 0.2.We evaluate the self-supervised learned representation through linear probing, i.e., we train a linear classifier on top of the encoder for 100 epochs and report its test accuracy.

## 5.1. Verification of the Trade-off between Two Distances

### 5.1.1. PIXEL-LEVEL VERIFICATION

We first conduct pixel-level verifications. Specifically, under certain augmentation parameters, for $n$ original input images, we first create $2n$ random augmented views. Note that to study the effect of a certain type of data augmentation, we vary its parameters (range of crop size and probability of color jitter) and set the default parameters for other types of augmentations. Then for each class, we calculate the minimum pixel-level distances between same-class (different) images and the maximum pixel-level distances between the two augmentations of the same image. We report the average maximum and minimum distances over all classes on CIFAR-100 and TinyImagenet in Figures 2 and 7.

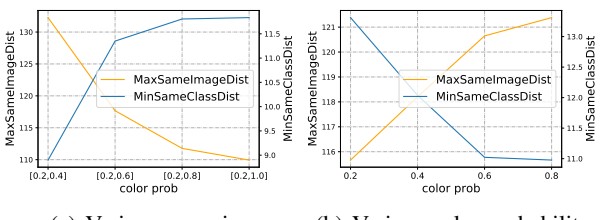

(a) Various crop size.  (b) Various color probability.

*Figure 2.* Pixel-level maximum distance between same-class different-image augmentations and minimum distance between same-image data augmentations on CIFAR-100.

In Figures 2 and 7, we show that as the augmentation strength increases (smaller range of crop size and higher probability of color jitter), the maximum distance between augmentations of the same image increases, whereas the minimum distance between same-class different images decreases. According to the theoretical discussions in Section 4, smaller crops reduce the minimum same-class distance by avoiding intersections with the semantic boundaries, and color distortion further reduces the distance for crops with pixels sharing the same semantic label. Moreover, smaller crops increase the maximum same-image distance by creating views with non-overlapping semantic labels, and color distortion further increases it by enhancing the bias term.

### 5.1.2. REPRESENTATION-LEVEL VERIFICATION

We also conduct empirical verification by calculating the distances in the embedding space. Specifically, we train SimCLR models with various parameters of data augmentation (range of crop size and probability of color jitter) and set the default parameters for other types of augmentations.

Then for each class, we calculate the minimum distances between same-class (different) images and the maximum distances between the two augmentations of the same image in the embedding space with training epoch from 1 to 1000. We report the average maximum and minimum distances over all classes on CIFAR-100 in Figures 3 and 4, and those on TinyImagenet in Figures 8 and 9. Note that for better visualization, we plot the moving average of the distance curves with a window=100 epochs.

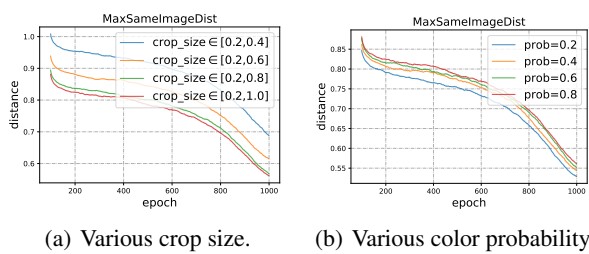

(a) Various crop size.  (b) Various color probability.

*Figure 3.* Representation-level maximum distance between same-class different-image augmentations on CIFAR-100.

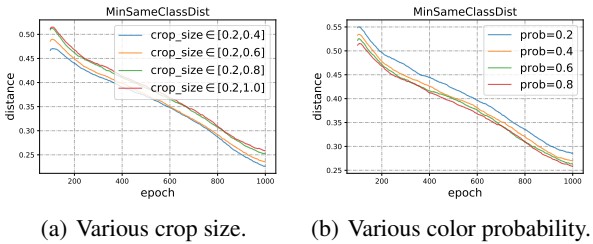

(a) Various crop size.  (b) Various color probability.

*Figure 4.* Representation-level minimum distance between different same-image data augmentations on CIFAR-100.

From Figures 3, 4, 8, and 9, we observe that through training, both distance values become smaller. Besides, at the beginning of the training stage, the rank of the representation-level distances w.r.t. augmentation parameters coincides with that of the pixel level, i.e., as the augmentation strength increases (smaller range of crop size and higher probability of color jitter), the maximum distance between augmentations of the same image increases, whereas the minimum distance between same-class different images decreases. Moreover, during training, this trend maintains till convergence.

## 5.2. Verification of Optimal Parameter

We run experiments with various augmentation parameters including the range of random crop size and the probability of random color jitter on the CIFAR-100 dataset. We report the sum of the two distance terms against training epochs in Figure 5. The curves are smoothed by taking the moving average over 100 epochs. Besides, in Figure 6, we show the linear probing accuracy of the unsupervised contrastive learning representations trained with various augmentation parameters. The results on TinyImagenet are shown in Figures 10 and 11.

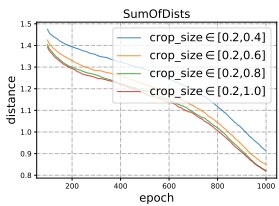 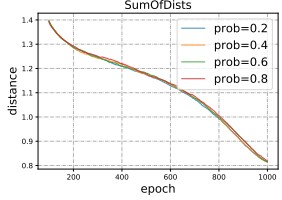

(a) Various crop size.    (b) Various color probability.

*Figure 5.* Sum of the two distance terms under various data augmentations in the embedding space on CIFAR-100.

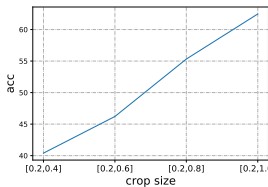 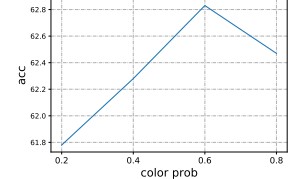

(a) Various crop size.    (b) Various color probability.

*Figure 6.* Linear probing accuracy under different data augmentation parameters on CIFAR-100.

According to Figures 5, 6, 10, and 11, we observe that the optimal augmentation parameter with the smallest distance sum also leads to the highest downstream accuracy. This verifies Theorem 2.3, which indicates that the downstream supervised risk is guaranteed by the sum of the maximum distance between same-class different-image augmentations and the minimum distance between different same-image data augmentations in the embedding space.

## 6. Conclusion

In this paper, by proposing an augmentation-aware error bound, we establish that the supervised risk is not only influenced by the unsupervised risk but also explicitly shaped by a trade-off induced by data augmentation. Under a novel semantic label assumption, we further analyze how specific augmentation methods impact this bound. Moreover, we empirically verify the theoretical conclusion on the trade-offs of data augmentation. We believe our study lays a foundation for further theoretically exploring data augmentation techniques in contrastive learning.

## Acknowledgements

Yisen Wang was supported by National Key R&D Program of China (2022ZD0160300), National Natural Science Foundation of China (92370129, 62376010), and Beijing Nova Program (20230484344, 20240484642).

## Impact Statement

This paper presents work whose goal is to advance the field of Machine Learning. There are many potential societal consequences of our work, none of which we feel must be specifically highlighted here.

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

## A. Related Works

### A.1. Self-Supervised Contrastive Learning

Self-supervised contrastive learning (Chen et al., 2020a;b; He et al., 2020; Chen et al., 2021) aims to train an encoder that maps different augmentations of the same input to similar feature representations, while ensuring that augmentations from distinct inputs result in distinct features. Once the encoder is pre-trained, it can be fine-tuned on a specific downstream task. Contrastive learning methods can be broadly divided into two categories based on their use of negative samples. The first category (Chen et al., 2020a;b; He et al., 2020) involves learning the encoder by aligning an anchor point with its augmented versions (positive samples) while explicitly pushing apart other samples (negative samples). The second category does not rely on negative samples and often incorporates additional components, such as projectors (Grill et al., 2020), stop-gradient techniques (Chen & He, 2021), or high-dimensional embeddings (Zbontar et al., 2021). Despite this, the first category remains the dominant approach in self-supervised contrastive learning and has been applied to a wide range of domains (Khaertdinov et al., 2021; Aberdam et al., 2021; Lee et al., 2022). This paper primarily analyzes and discusses the first category of contrastive learning methods, which depend on both positive and negative samples.

### A.2. Theory of Contrastive Learning

The early studies of theoretical aspects of contrastive learning manage to link contrastive learning to the supervised downstream classification. Arora et al. (2019) first proposed the CURL framework where the positive samples are generated from the same latent class. They proved that the downstream supervised risk of a mean classifier can be bounded by the unsupervised contrastive risk and a class collision term. Under the CURL framework, succeeding studies further extended this bound and incorporated the effect of negative samples (Ash et al., 2022; Bao et al., 2022). Moreover, Lei et al. (2023) improved the results of Arora et al. (2019) by deriving a tighter generalization bound. However, the CURL framework has long been criticized as having unrealistic generation process of positive samples (Wang et al., 2021). Nozawa & Sato (2021) formulated the positive samples from the perspective of data augmentations, whereas they considered the corresponding term as a relative constant without conducting further analysis.

Later on, HaoChen et al. (2021) motivated from the unsupervised nature of contrastive learning by proposing the concept of the augmentation graph, where the vertices are all augmented images and the edge weights representing the probability of two augmented views originating from the same original image. They borrowed the mathematical tools from spectral clustering to build generalization guarantees for their proposed spectral contrastive learning. The theoretical framework was later extended to contrastive learning for unsupervised domain adaption (Shen et al., 2022), multi-modal learning (Zhang et al., 2023a), and weakly supervised learning (Cui et al., 2023). In a similar vein, Wang et al. (2021) propose the idea of augmentation overlap to explain the alignment of positive samples. However, these modelings of data augmentation measure the case where two original images sharing the same view, whereas this is empirically hard to realize, especially with only two views used in contrastive learning instead of multiple ones.

Besides, contrastive learning is also interpreted through various other theoretical frameworks in unsupervised learning. For example, Wang & Isola (2020) explained the contrastive learning through alignment of positive samples and uniformity of negative samples. Zimmermann et al. (2021) showed that training with InfoNCE inverts the data generating process through establishing a theoretical connection between InfoNCE and nonlinear independent component analysis (ICA). Ko et al. (2022) built the relationship between contrastive learning and neighborhood component analysis (NCA) and developed new contrastive losses. Hu et al. (2023) interpreted contrastive learning as a type of stochastic neighbor embedding (SNE) methods. Wang et al. (2023) showed the learning dynamics of contrastive learning corresponds to a specific message passing scheme on the corresponding augmentation graph. Wu et al. (2024) explained the tolerance of contrastive learning towards sampling bias via the perspective of distributionally robust optimization (DRO). Nonetheless, the role of data augmentation is still under-exploited in the existing theoretical frameworks, especially without mathematically analyzing specific data augmentation methods.

## B. Proofs

### B.1. Proof of Error Decomposition

*Proof of Theorem 2.5.* According to the definition, we have

$$\mathcal{R}^{\mathrm{un}}(f) = \mathbb{E}_{c,\{c_k\}_{k\in[K]}} \mathbb{E}_{\bar{x}\sim\rho_c,\bar{x}_k\sim\rho_{c_k}} \mathbb{E}_{a,a',\{a_k\}_{k\in[K]}} \mathcal{L}(a(\bar{x}), a'(\bar{x}), a_k(\bar{x}_k); f)$$

$$= \mathbb{E}_c \mathbb{E}_{\bar{x} \sim \rho_c} \mathbb{E}_a \sum_{i_1=1}^{C} \cdots \sum_{i_K=1}^{C} \mathrm{P}(c_1 = i_1) \cdots \mathrm{P}(c_K = i_K) \cdot \mathbb{E}_{\bar{x}_k \sim \rho_{i_k}} \mathbb{E}_{a', \{a_k\}_{k \in [K]}} \mathcal{L}(a(\bar{x}), a'(\bar{x}), a_k(\bar{x}_k); f)$$

$$= \mathbb{E}_c \mathbb{E}_{\bar{x} \sim \rho_c} \mathbb{E}_a \Big( \sum_{i_1 \neq c} + \sum_{i_1 = c} \Big) \cdots \Big( \sum_{i_K \neq c} + \sum_{i_K = c} \Big) \mathrm{P}(c_1 = i_1) \cdots \mathrm{P}(c_K = i_K)$$
$$\cdot \mathbb{E}_{\bar{x}_k \sim \rho_{i_k}} \mathbb{E}_{a', \{a_k\}_{k \in [K]}} \mathcal{L}(a(\bar{x}), a'(\bar{x}), a_k(\bar{x}_k); f)$$

$$= \mathbb{E}_c \mathbb{E}_{\bar{x} \sim \rho_c} \mathbb{E}_a \sum_{i_1 \neq c} \cdots \sum_{i_K \neq c} \mathrm{P}(c_1 = i_1) \cdots \mathrm{P}(c_K = i_K) \cdot \mathbb{E}_{\bar{x}_k \sim \rho_{i_k}, k \in [K]} \mathbb{E}_{a', \{a_k\}_{k \in [K]}} \mathcal{L}(a(\bar{x}), a'(\bar{x}), a_k(\bar{x}_k); f)$$

$$+ \mathbb{E}_c \mathbb{E}_{\bar{x} \sim \rho_c} \mathbb{E}_a \sum_{j=1}^{K} \sum_{i_1 \neq c} \cdots \sum_{i_{j-1} \neq c} \cdot \sum_{i_{j+1} \neq c} \cdots \sum_{i_K \neq c} \mathrm{P}(c_1 = i_1) \cdots \mathrm{P}(c_j = c) \cdots \mathrm{P}(c_K = i_K)$$
$$\cdot \mathbb{E}_{\bar{x}_j \sim \rho_c} \mathbb{E}_{\bar{x}_k \sim \rho_{i_k}, k \neq j} \mathbb{E}_{a', \{a_k\}_{k \in [K]}} \mathcal{L}(a(\bar{x}), a'(\bar{x}), a_k(\bar{x}_k); f)$$

$$\cdots$$

$$+ \mathbb{E}_c \mathbb{E}_{\bar{x} \sim \rho_c} \mathbb{E}_a \sum_{j=1}^{K} \sum_{i_j \neq c} \mathrm{P}(c_1 = c) \cdots \mathrm{P}(c_j = i_j) \cdots \mathrm{P}(c_K = c)$$
$$\cdot \mathbb{E}_{\bar{x}_k \sim \rho_c, k \neq j} \mathbb{E}_{\bar{x}_j \sim \rho_{i_j}} \mathbb{E}_{a', \{a_k\}_{k \in [K]}} \mathcal{L}(a(\bar{x}), a'(\bar{x}), a_k(\bar{x}_k); f)$$
$$+ \mathbb{E}_c \mathbb{E}_{\bar{x} \sim \rho_c} \mathbb{E}_a \mathrm{P}(c_1 = c) \cdots \mathrm{P}(c_K = c) \cdot \mathbb{E}_{\bar{x}_k \sim \rho_c, k \in [K]} \mathbb{E}_{a', \{a_k\}_{k \in [K]}} \mathcal{L}(a(\bar{x}), a'(\bar{x}), a_k(\bar{x}_k); f)$$

$$= \mathbb{E}_c \mathbb{E}_{\bar{x} \sim \rho_c} \mathbb{E}_a \sum_{i_1 \neq c} \cdots \sum_{i_K \neq c} \mathrm{P}(c_1 = i_1) \cdots \mathrm{P}(c_K = i_K) \cdot \mathbb{E}_{\bar{x}_k \sim \rho_{i_k}, k \in [K]} \mathbb{E}_{a', \{a_k\}_{k \in [K]}} \mathcal{L}(a(\bar{x}), a'(\bar{x}), a_k(\bar{x}_k); f)$$

$$+ \mathbb{E}_c \mathbb{E}_{\bar{x} \sim \rho_c} \mathbb{E}_a \sum_{i_1 \neq c} \cdots \sum_{i_{j-1} \neq c} \cdot \sum_{i_{j+1} \neq c} \cdots \sum_{i_K \neq c} \sum_{j=1}^{K} \mathrm{P}(c_1 = i_1) \cdots \mathrm{P}(c_j = c) \cdots \mathrm{P}(c_K = i_K)$$
$$\cdot \mathbb{E}_{\bar{x}_j \sim \rho_c} \mathbb{E}_{\bar{x}_k \sim \rho_{i_k}, k \neq j} \mathbb{E}_{a', \{a_k\}_{k \in [K]}} \mathcal{L}(a(\bar{x}), a'(\bar{x}), a_k(\bar{x}_k); f)$$

$$\cdots$$

$$+ \mathbb{E}_c \mathbb{E}_{\bar{x} \sim \rho_c} \mathbb{E}_a \sum_{i_j \neq c} \sum_{j=1}^{K} \mathrm{P}(c_1 = c) \cdots \mathrm{P}(c_j = i_j) \cdots \mathrm{P}(c_K = c)$$
$$\cdot \mathbb{E}_{\bar{x}_k \sim \rho_c, k \neq j} \mathbb{E}_{\bar{x}_j \sim \rho_{i_j}} \mathbb{E}_{a', \{a_k\}_{k \in [K]}} \mathcal{L}(a(\bar{x}), a'(\bar{x}), a_k(\bar{x}_k); f)$$
$$+ \mathbb{E}_c \mathbb{E}_{\bar{x} \sim \rho_c} \mathbb{E}_a \mathrm{P}(c_1 = c) \cdots \mathrm{P}(c_K = c) \cdot \mathbb{E}_{\bar{x}_k \sim \rho_c, k \in [K]} \mathbb{E}_{a', \{a_k\}_{k \in [K]}} \mathcal{L}(a(\bar{x}), a'(\bar{x}), a_k(\bar{x}_k); f). \tag{26}$$

Note that the loss $\mathcal{L}(a(\bar{x}), a'(\bar{x}), a_k(\bar{x}_k); f)$ is symmetric w.r.t. negative samples. Specifically, denoting $\{i_1, \ldots, i_K\}$ as a rearrangement of $[K] = \{1, \ldots, K\}$, we have

$$\mathcal{L}(a(\bar{x}), a'(\bar{x}), a_k(\bar{x}_k); f) = \log \Big( 1 + \sum_{k=1}^{K} \exp \big( - f(a(\bar{x}))^\top [f(a'(\bar{x})) - f(a_k(\bar{x}_k))] \big) \Big)$$
$$= \log \Big( 1 + \sum_{k=i_1}^{i_K} \exp \big( - f(a(\bar{x}))^\top [f(a'(\bar{x})) - f(a_k(\bar{x}_k))] \big) \Big). \tag{27}$$

Therefore, for $k \in [K]$, given $i_1, \ldots, i_k$, by denoting

$$r_k(i_1, \ldots, i_k) := \mathbb{E}_{\bar{x}_1 \sim \rho_{i_1}} \cdots \mathbb{E}_{\bar{x}_k \sim \rho_{i_k}} \mathbb{E}_{\bar{x}_{k+1}, \ldots, \bar{x}_K \sim \rho_c} \mathbb{E}_{a', \{a_k\}_{k \in [K]}} \mathcal{L}(a(\bar{x}), a'(\bar{x}), a_k(\bar{x}_k); f), \tag{28}$$

and

$$p_k(i_1, \ldots, i_k) := \mathrm{P}(\exists \{j_1, \ldots, j_K\}, \text{such that } c_{j_1} = i_1, \ldots, c_{j_k} = i_k, \text{and } c_{j_{k+1} = \cdots = c_K = c}), \tag{29}$$

where $\{j_1, \ldots, j_K\}$ is a rearrangement of $[K] = \{1, \ldots, K\}$, we have

$$\mathcal{R}^{\mathrm{un}}(f) = \mathbb{E}_c \mathbb{E}_{\bar{x} \sim \rho_c} \mathbb{E}_a \sum_{i_1 \neq c} \cdots \sum_{i_K \neq c} p_K(i_1, \ldots, i_K) r_K(i_1, \ldots, i_K)$$

$$+ \mathbb{E}_c \mathbb{E}_{\bar{x} \sim \rho_c} \mathbb{E}_a \sum_{i_1 \neq c} \cdots \sum_{i_{j-1} \neq c} \cdot \sum_{i_{j+1} \neq c} \cdots \sum_{i_K \neq c} p_{K-1}(i_1, \ldots, i_{j-1}, i_{j+1} \ldots i_K) r_{K-1}(i_1, \ldots, i_{j-1}, i_{j+1} \ldots i_K)$$

$$\cdots$$

$$+ \mathbb{E}_c \mathbb{E}_{\bar{x} \sim \rho_c} \mathbb{E}_a \sum_{i_j \neq c} p_1(i_j) r_1(i_j) + \mathbb{E}_c \mathbb{E}_{\bar{x} \sim \rho_c} \mathbb{E}_a p_0 r_0. \tag{30}$$

$\square$

**Corollary B.1** (Error decomposition of $\bar{\mathcal{R}}^{\mathrm{sup}}$). *We have*

$$\bar{\mathcal{R}}^{\mathrm{sup}}(f)$$

$$= \mathbb{E}_c \mathbb{E}_{\bar{x} \sim \rho_c} \sum_{i_1 \neq c} \cdots \sum_{i_K \neq c} p_K(i_1, \ldots, i_K) r_K^{\mathrm{sup}}(i_1, \ldots, i_K)$$

$$\cdots$$

$$+ \mathbb{E}_c \mathbb{E}_{\bar{x} \sim \rho_c} \sum_{i_j \neq c} p_1(i_j) r_1^{\mathrm{sup}}(i_j) + \mathbb{E}_c \mathbb{E}_{\bar{x} \sim \rho_c} p_0 r_0^{\mathrm{sup}}, \tag{31}$$

*where* $r_k^{\mathrm{sup}}(i_1, \ldots, i_k) := \log \left( 1 + \left[ (K - k) + \sum_{m=1}^k \exp \left( - \mathbb{E}_{\bar{x}' \sim \rho_c} \mathbb{E}_{\bar{x}_m \sim \rho_{i_m}} f(\bar{x})^\top [f(\bar{x}') - f(\bar{x}_m)] \right) \right] \right).$

*Proof of Corollary B.1.* Because $\log \left( 1 + \sum_{k=1}^K \exp \left( - f(\bar{x})^\top (\mu_c - \mu_{c_k}) \right) \right)$ is symmetric w.r.t. the negative samples, we follow the proof of Theorem 2.5, replace $r_k(i_1, \ldots, i_k)$ with $r_k^{\mathrm{sup}}(i_1, \ldots, i_k)$, and finish the proof. $\square$

### B.2. Proof of the Error Bound without Additional Assumption

*Proof of Theorem 2.6.* By Jensen's inequality and the convexity of log-sum-exp, we have

$$r_k(i_1, \ldots, i_k)$$

$$= \mathbb{E}_{\bar{x}_1 \sim \rho_{i_1}} \cdots \mathbb{E}_{\bar{x}_k \sim \rho_{i_k}} \mathbb{E}_{\bar{x}_{k+1}, \ldots, \bar{x}_K \sim \rho_c} \mathbb{E}_{a', \{a_k\}_{k \in [K]}} \mathcal{L}(a(\bar{x}), a'(\bar{x}), a_k(\bar{x}_k); f)$$

$$= \mathbb{E}_{\bar{x}_1 \sim \rho_{i_1}} \cdots \mathbb{E}_{\bar{x}_k \sim \rho_{i_k}} \mathbb{E}_{\bar{x}_{k+1}, \ldots, \bar{x}_K \sim \rho_c} \mathbb{E}_{a', \{a_k\}_{k \in [K]}} \log \left( 1 + \sum_{k=1}^K \exp \left( - f(a(\bar{x}))^\top [f(a'(\bar{x})) - f(a_k(\bar{x}_k))] \right) \right)$$

$$\geq \log \left( 1 + \sum_{k=1}^K \exp \left( - \mathbb{E}_{\bar{x}_1 \sim \rho_{i_1}} \cdots \mathbb{E}_{\bar{x}_k \sim \rho_{i_k}} \mathbb{E}_{\bar{x}_{k+1}, \ldots, \bar{x}_K \sim \rho_c} \mathbb{E}_{a', \{a_k\}_{k \in [K]}} f(a(\bar{x}))^\top [f(a'(\bar{x})) - f(a_k(\bar{x}_k))] \right) \right)$$

$$= \log \left( 1 + \sum_{m=1}^k \exp \left( - \mathbb{E}_{\bar{x}_m \sim \rho_{i_m}} \mathbb{E}_{a', a_m} f(a(\bar{x}))^\top [f(a'(\bar{x})) - f(a_m(\bar{x}_m))] \right) \right.$$

$$\left. + \sum_{m=k+1}^K \exp \left( - \mathbb{E}_{\bar{x}_m \sim \rho_c} \mathbb{E}_{a', a_m} f(a(\bar{x}))^\top [f(a'(\bar{x})) - f(a_m(\bar{x}_m))] \right) \right). \tag{32}$$

We note that the negative samples break into two groups: $\bar{x}_m$'s sharing the same class with $\bar{x}$, and $\bar{x}_m$'s having different classes with $\bar{x}$. For the same-class terms, given $a$, $\bar{x}$, $\bar{x}_m$, and $f$, we take $a^*$ as the augmentation such that

$$a^* = \arg \min_a \|f(a_1(\bar{x})) - f(a(\bar{x}_k))\|, \tag{33}$$

and then we have

$$\mathbb{E}_{\bar{x}_m \sim \rho_c} \mathbb{E}_{a', a_m} f(a(\bar{x}))^\top [f(a'(\bar{x})) - f(a_m(\bar{x}_m))]$$

$$= \mathbb{E}_{\bar{x}_m \sim \rho_c} \mathbb{E}_{a', a_m} f(a(\bar{x}))^\top [f(a'(\bar{x})) - f(a^*(\bar{x}_m)) + f(a^*(\bar{x}_m)) - f(a_m(\bar{x}_m))]$$

$$= \mathbb{E}_{\bar{x}_m \sim \rho_c} \mathbb{E}_{a'} f(a(\bar{x}))^\top [f(a'(\bar{x})) - f(a^*(\bar{x}_m))] + \mathbb{E}_{\bar{x}_m \sim \rho_c} \mathbb{E}_{a_m} f(a(\bar{x}))^\top [f(a^*(\bar{x}_m)) - f(a_m(\bar{x}_m))]$$

$$\leq \mathbb{E}_{\bar{x}_m \sim \rho_c} \mathbb{E}_{a'} \|f(a'(\bar{x})) - f(a^*(\bar{x}_m))\| + \mathbb{E}_{\bar{x}_m \sim \rho_c} \mathbb{E}_{a_m} \|f(a^*(\bar{x}_m)) - f(a_m(\bar{x}_m))\|$$

$$\leq \mathbb{E}_{\bar{x}_m \sim \rho_c} \mathbb{E}_{a'} \min_{a''} \|f(a'(\bar{x})) - f(a''(\bar{x}_m))\| + \mathbb{E}_{\bar{x}_m \sim \rho_c} \max_{a',a''} \|f(a''(\bar{x}_m)) - f(a'(\bar{x}_m))\|, \tag{34}$$

where the first inequality holds because $\|f(\cdot)\| = 1$.

On the other hand, for the different-class terms, we have

$$\mathbb{E}_{\bar{x}_m \sim \rho_{i_m}} \mathbb{E}_{a',a_m} f(a(\bar{x}))^\top [f(a'(\bar{x})) - f(a_m(\bar{x}_m))]$$
$$= \mathbb{E}_{\bar{x}' \sim \rho_c} \mathbb{E}_{\bar{x}_m \sim \rho_{i_m}} \mathbb{E}_{a',a'',a_m} f(a(\bar{x}))^\top [f(a'(\bar{x})) - f(a''(\bar{x}')) + f(a''(\bar{x}')) - f(a_m(\bar{x}_m))]$$
$$= \mathbb{E}_{\bar{x}' \sim \rho_c} \mathbb{E}_{a',a''} f(a(\bar{x}))^\top [f(a'(\bar{x})) - f(a''(\bar{x}'))] + \mathbb{E}_{\bar{x}' \sim \rho_c} \mathbb{E}_{\bar{x}_m \sim \rho_{i_m}} \mathbb{E}_{a'',a_m} f(a(\bar{x}))^\top [f(a''(\bar{x}')) - f(a_m(\bar{x}_m))]. \tag{35}$$

Following (34), the first term of (35) is bounded by

$$\mathbb{E}_{\bar{x}' \sim \rho_c} \mathbb{E}_{a',a''} f(a(\bar{x}))^\top [f(a'(\bar{x})) - f(a''(\bar{x}'))]$$
$$\leq \mathbb{E}_{\bar{x}' \sim \rho_c} \mathbb{E}_{a'} \min_{a''} \|f(a'(\bar{x})) - f(a''(\bar{x}'))\| + \mathbb{E}_{\bar{x}' \sim \rho_c} \max_{a',a''} \|f(a''(\bar{x}')) - f(a'(\bar{x}'))\|. \tag{36}$$

Besides, we decompose the second term of (35) as follows.

$$\mathbb{E}_{\bar{x}' \sim \rho_c} \mathbb{E}_{\bar{x}_m \sim \rho_{i_m}} \mathbb{E}_{a'',a_m} f(a(\bar{x}))^\top [f(a''(\bar{x}')) - f(a_m(\bar{x}_m))]$$
$$= \mathbb{E}_{\bar{x}' \sim \rho_c} \mathbb{E}_{\bar{x}_m \sim \rho_{i_m}} f(\bar{x})^\top [f(\bar{x}') - f(\bar{x}_m)]$$
$$+ \mathbb{E}_{\bar{x}' \sim \rho_c} \mathbb{E}_{\bar{x}_m \sim \rho_{i_m}} \mathbb{E}_{a'',a_m} [f(a(\bar{x})) - f(\bar{x})]^\top [f(a''(\bar{x}')) - f(a_m(\bar{x}_m))]$$
$$+ \mathbb{E}_{\bar{x}' \sim \rho_c} \mathbb{E}_{a''} f(\bar{x})^\top [f(a''(\bar{x}')) - f(\bar{x}')] + \mathbb{E}_{\bar{x}_m \sim \rho_{i_m}} \mathbb{E}_{a_m} f(\bar{x})^\top [f(\bar{x}_m) - f(a_m(\bar{x}_m))]$$
$$\leq \mathbb{E}_{\bar{x}' \sim \rho_c} \mathbb{E}_{\bar{x}_m \sim \rho_{i_m}} f(\bar{x})^\top [f(\bar{x}') - f(\bar{x}_m)] + 2\|f(a(\bar{x})) - f(\bar{x})\|$$
$$+ \mathbb{E}_{\bar{x}' \sim \rho_c} \mathbb{E}_{a''} \|f(a''(\bar{x}')) - f(\bar{x}')\| + \mathbb{E}_{\bar{x}_m \sim \rho_{i_m}} \mathbb{E}_{a_m} \|f(\bar{x}_m) - f(a_m(\bar{x}_m))\|$$
$$= \mathbb{E}_{\bar{x}' \sim \rho_c} \mathbb{E}_{\bar{x}_m \sim \rho_{i_m}} f(\bar{x})^\top [f(\bar{x}') - f(\bar{x}_m)] + 2\|f(a(\bar{x})) - f(\bar{x})\|$$
$$+ \mathbb{E}_{\bar{x}' \sim \rho_c} \mathbb{E}_{a''} \|f(a''(\bar{x}')) - f(\mathrm{Id}(\bar{x}'))\| + \mathbb{E}_{\bar{x}_m \sim \rho_{i_m}} \mathbb{E}_{a_m} \|f(\mathrm{Id}(\bar{x}_m)) - f(a_m(\bar{x}_m))\|$$
$$\leq \mathbb{E}_{\bar{x}' \sim \rho_c} \mathbb{E}_{\bar{x}_m \sim \rho_{i_m}} f(\bar{x})^\top [f(\bar{x}') - f(\bar{x}_m)] + 2\|f(a(\bar{x})) - f(\bar{x})\|$$
$$+ \mathbb{E}_{\bar{x}' \sim \rho_c} \max_{a',a''} \|f(a''(\bar{x}')) - f(a'(\bar{x}'))\| + \mathbb{E}_{\bar{x}_m \sim \rho_{i_m}} \max_{a',a''} \|f(a''(\bar{x}_m)) - f(a'(\bar{x}_m))\|, \tag{37}$$

where the last inequality holds if $\mathrm{Id}(\cdot) \in \{a : a \sim \mathcal{A}\}$. Combining (36) and (37), we have the different-class terms bounded by

$$\mathbb{E}_{\bar{x}_m \sim \rho_{i_m}} \mathbb{E}_{a',a_m} f(a(\bar{x}))^\top [f(a'(\bar{x})) - f(a_m(\bar{x}_m))]$$
$$\leq \mathbb{E}_{\bar{x}' \sim \rho_c} \mathbb{E}_{a'} \min_{a''} \|f(a'(\bar{x})) - f(a''(\bar{x}'))\| + \mathbb{E}_{\bar{x}' \sim \rho_c} \max_{a',a''} \|f(a''(\bar{x}')) - f(a'(\bar{x}'))\|$$
$$+ \mathbb{E}_{\bar{x}' \sim \rho_c} \mathbb{E}_{\bar{x}_m \sim \rho_{i_m}} f(\bar{x})^\top [f(\bar{x}') - f(\bar{x}_m)] + 2\|f(a(\bar{x})) - f(\bar{x})\|$$
$$+ \mathbb{E}_{\bar{x}' \sim \rho_c} \max_{a',a''} \|f(a''(\bar{x}')) - f(a'(\bar{x}'))\| + \mathbb{E}_{\bar{x}_m \sim \rho_{i_m}} \max_{a',a''} \|f(a''(\bar{x}_m)) - f(a'(\bar{x}_m))\|$$
$$= \mathbb{E}_{\bar{x}' \sim \rho_c} \mathbb{E}_{\bar{x}_m \sim \rho_{i_m}} f(\bar{x})^\top [f(\bar{x}') - f(\bar{x}_m)] + \mathbb{E}_{\bar{x}' \sim \rho_c} \mathbb{E}_{a'} \min_{a''} \|f(a'(\bar{x})) - f(a''(\bar{x}'))\|$$
$$+ 2\|f(a(\bar{x})) - f(\bar{x})\| + 2\mathbb{E}_{\bar{x}' \sim \rho_c} \max_{a',a''} \|f(a''(\bar{x}')) - f(a'(\bar{x}'))\| + \mathbb{E}_{\bar{x}_m \sim \rho_{i_m}} \max_{a',a''} \|f(a''(\bar{x}_m)) - f(a'(\bar{x}_m))\|. \tag{38}$$

Then plugging (34) and (38) into (32), we have

$$r_k(i_1, \ldots, i_k)$$
$$\geq \log \Big( 1 + \sum_{m=1}^{k} \exp \big( - [\mathbb{E}_{\bar{x}' \sim \rho_c} \mathbb{E}_{\bar{x}_m \sim \rho_{i_m}} f(\bar{x})^\top [f(\bar{x}') - f(\bar{x}_m)] + \mathbb{E}_{\bar{x}' \sim \rho_c} \mathbb{E}_{a'} \min_{a''} \|f(a'(\bar{x})) - f(a''(\bar{x}'))\|$$
$$+ 2\|f(a(\bar{x})) - f(\bar{x})\| + 2\mathbb{E}_{\bar{x}' \sim \rho_c} \max_{a',a''} \|f(a''(\bar{x}')) - f(a'(\bar{x}'))\| + \mathbb{E}_{\bar{x}_m \sim \rho_{i_m}} \max_{a',a''} \|f(a''(\bar{x}_m)) - f(a'(\bar{x}_m))\|] \big)$$
$$+ (K - k) \exp \big( - [\mathbb{E}_{\bar{x}_m \sim \rho_c} \mathbb{E}_{a'} \min_{a''} \|f(a'(\bar{x})) - f(a''(\bar{x}_m))\| + \mathbb{E}_{\bar{x}_m \sim \rho_c} \max_{a',a''} \|f(a''(\bar{x}_m)) - f(a'(\bar{x}_m))\|] \big) \Big)$$

$$\geq \log\Big(1 + \sum_{m=1}^{k} \exp\big(-\mathbb{E}_{\bar{x}'\sim\rho_c}\mathbb{E}_{\bar{x}_m\sim\rho_{i_m}}f(\bar{x})^{\top}[f(\bar{x}') - f(\bar{x}_m)]\big) \cdot \exp\big(-[\mathbb{E}_{\bar{x}'\sim\rho_c}\mathbb{E}_{a'}\min_{a''}\|f(a'(\bar{x})) - f(a''(\bar{x}'))\|$$

$$+ 2\|f(a(\bar{x})) - f(\bar{x})\| + 2\mathbb{E}_{\bar{x}'\sim\rho_c}\max_{a',a''}\|f(a''(\bar{x}')) - f(a'(\bar{x}'))\| + \mathbb{E}_{\bar{x}_m\sim\rho_{i_m}}\max_{a',a''}\|f(a''(\bar{x}_m)) - f(a'(\bar{x}_m))\|]\big)$$

$$+ (K-k)\exp\big(-[\mathbb{E}_{\bar{x}'\sim\rho_c}\mathbb{E}_{a'}\min_{a''}\|f(a'(\bar{x})) - f(a''(\bar{x}'))\| + \mathbb{E}_{\bar{x}'\sim\rho_c}\max_{a',a''}\|f(a''(\bar{x}')) - f(a'(\bar{x}'))\|]\big)\Big)$$

$$= \log\Big(1 + \big[(K-k) + \sum_{m=1}^{k}\exp\big(-\mathbb{E}_{\bar{x}'\sim\rho_c}\mathbb{E}_{\bar{x}_m\sim\rho_{i_m}}f(\bar{x})^{\top}[f(\bar{x}') - f(\bar{x}_m)]\big)\cdot\exp\big(-[2\|f(a(\bar{x})) - f(\bar{x})\|$$

$$+ \mathbb{E}_{\bar{x}'\sim\rho_c}\max_{a',a''}\|f(a''(\bar{x}')) - f(a'(\bar{x}'))\| + \mathbb{E}_{\bar{x}_m\sim\rho_{i_m}}\max_{a',a''}\|f(a''(\bar{x}_m)) - f(a'(\bar{x}_m))\|]\big)\big]$$

$$\cdot \exp\big(-[\mathbb{E}_{\bar{x}'\sim\rho_c}\mathbb{E}_{a'}\min_{a''}\|f(a'(\bar{x})) - f(a''(\bar{x}'))\| + \mathbb{E}_{\bar{x}'\sim\rho_c}\max_{a',a''}\|f(a''(\bar{x}')) - f(a'(\bar{x}'))\|]\big)\Big)$$

$$\geq \log\Big(1 + \big[(K-k) + \sum_{m=1}^{k}\exp\big(-\mathbb{E}_{\bar{x}'\sim\rho_c}\mathbb{E}_{\bar{x}_m\sim\rho_{i_m}}f(\bar{x})^{\top}[f(\bar{x}') - f(\bar{x}_m)]\big)\cdot\exp\big(-[2\|f(a(\bar{x})) - f(\bar{x})\|$$

$$+ \mathbb{E}_{\bar{x}'\sim\rho_c}\max_{a',a''}\|f(a''(\bar{x}')) - f(a'(\bar{x}'))\| + \mathbb{E}_{\bar{x}_m\sim\rho_{i_m}}\max_{a',a''}\|f(a''(\bar{x}_m)) - f(a'(\bar{x}_m))\|]\big)\big]\Big)$$

$$- [\mathbb{E}_{\bar{x}'\sim\rho_c}\mathbb{E}_{a'}\min_{a''}\|f(a'(\bar{x})) - f(a''(\bar{x}'))\| + \mathbb{E}_{\bar{x}'\sim\rho_c}\max_{a',a''}\|f(a''(\bar{x}')) - f(a'(\bar{x}'))\|]$$

$$\geq \log\Big(1 + \big[(K-k) + \sum_{m=1}^{k}\exp\big(-\mathbb{E}_{\bar{x}'\sim\rho_c}\mathbb{E}_{\bar{x}_m\sim\rho_{i_m}}f(\bar{x})^{\top}[f(\bar{x}') - f(\bar{x}_m)]\big)\big]\Big)$$

$$- [2\|f(a(\bar{x})) - f(\bar{x})\| + \mathbb{E}_{\bar{x}'\sim\rho_c}\max_{a',a''}\|f(a''(\bar{x}')) - f(a'(\bar{x}'))\| + \mathbb{E}_{\bar{x}_m\sim\rho_{i_m}}\max_{a',a''}\|f(a''(\bar{x}_m)) - f(a'(\bar{x}_m))\|]$$

$$= \log\Big(1 + \big[(K-k) + \sum_{m=1}^{k}\exp\big(-\mathbb{E}_{\bar{x}'\sim\rho_c}\mathbb{E}_{\bar{x}_m\sim\rho_{i_m}}f(\bar{x})^{\top}[f(\bar{x}') - f(\bar{x}_m)]\big)\big]\Big)$$

$$- [2\|f(a(\bar{x})) - f(\bar{x})\| + 2\mathbb{E}_{\bar{x}'\sim\rho_c}\max_{a',a''}\|f(a''(\bar{x}')) - f(a'(\bar{x}'))\| + \mathbb{E}_{\bar{x}_m\sim\rho_{i_m}}\max_{a',a''}\|f(a''(\bar{x}_m)) - f(a'(\bar{x}_m))\|$$

$$+ \mathbb{E}_{\bar{x}'\sim\rho_c}\mathbb{E}_{a'}\min_{a''}\|f(a'(\bar{x})) - f(a''(\bar{x}'))\|]$$

$$:= r_k^{\sup}(i_1,\ldots,i_k) - [2\|f(a(\bar{x})) - f(\bar{x})\| + 2\mathbb{E}_{\bar{x}'\sim\rho_c}\max_{a',a''}\|f(a''(\bar{x}')) - f(a'(\bar{x}'))\|$$

$$+ \mathbb{E}_{\bar{x}_m\sim\rho_{i_m}}\max_{a',a''}\|f(a''(\bar{x}_m)) - f(a'(\bar{x}_m))\| + \mathbb{E}_{\bar{x}'\sim\rho_c}\mathbb{E}_{a'}\min_{a''}\|f(a'(\bar{x})) - f(a''(\bar{x}'))\|]. \tag{39}$$

$\square$

*Proof of Theorem 2.7.* Combining Theorems 2.5 and 2.6, we have

$$\mathcal{R}^{\mathrm{un}}(f) = \mathbb{E}_c\mathbb{E}_{\bar{x}\sim\rho_c}\mathbb{E}_a \sum_{i_1\neq c}\cdots\sum_{i_K\neq c} p_K(i_1,\ldots,i_K)r_K(i_1,\ldots,i_K)$$

$$+ \mathbb{E}_c\mathbb{E}_{\bar{x}\sim\rho_c}\mathbb{E}_a \sum_{i_1\neq c}\cdots\sum_{i_{j-1}\neq c}\cdot\sum_{i_{j+1}\neq c}\cdots\sum_{i_K\neq c} p_{K-1}(i_1,\ldots,i_{j-1},i_{j+1}\ldots i_K)r_{K-1}(i_1,\ldots,i_{j-1},i_{j+1}\ldots i_K)$$

$$\cdots$$

$$+ \mathbb{E}_c\mathbb{E}_{\bar{x}\sim\rho_c}\mathbb{E}_a \sum_{i_j\neq c} p_1(i_j)r_1(i_j) + \mathbb{E}_c\mathbb{E}_{\bar{x}\sim\rho_c}\mathbb{E}_a p_0 r_0$$

$$\geq \mathbb{E}_c\mathbb{E}_{\bar{x}\sim\rho_c} \sum_{i_1\neq c}\cdots\sum_{i_K\neq c} p_K(i_1,\ldots,i_K)r_K^{\sup}(i_1,\ldots,i_K)$$

$$+ \mathbb{E}_c\mathbb{E}_{\bar{x}\sim\rho_c} \sum_{i_1\neq c}\cdots\sum_{i_{j-1}\neq c}\cdot\sum_{i_{j+1}\neq c}\cdots\sum_{i_K\neq c} p_{K-1}(i_1,\ldots,i_{j-1},i_{j+1}\ldots i_K)r_{K-1}^{\sup}(i_1,\ldots,i_{j-1},i_{j+1}\ldots i_K)$$

$$\cdots$$

$$+ \mathbb{E}_c \mathbb{E}_{\bar{x} \sim \rho_c} \sum_{i_j \neq c} p_1(i_j) r_1^{\sup}(i_j) + \mathbb{E}_c \mathbb{E}_{\bar{x} \sim \rho_c} p_0 r_0^{\sup}$$

$$- \mathbb{E}_c \mathbb{E}_{\bar{x} \sim \rho_c} \mathbb{E}_a [2\|f(a(\bar{x})) - f(\bar{x})\| + 2\mathbb{E}_{\bar{x}' \sim \rho_c} \max_{a', a''} \|f(a''(\bar{x}')) - f(a'(\bar{x}'))\| + \mathbb{E}_{\bar{x}' \sim \rho_c} \mathbb{E}_{a'} \min_{a''} \|f(a'(\bar{x})) - f(a''(\bar{x}'))\|]$$

$$- \mathbb{E}_c \sum_{k=0}^{K} \sum_{\{j_1, \ldots, j_K\} \in \mathrm{re}([K])} \sum_{i_1 \neq c} \cdots \sum_{i_k \neq c} \mathrm{P}(c_{j_1} = i_1) \ldots \mathrm{P}(c_{j_k} = i_k) \mathrm{P}(c_{j_{k+1}} = c) \ldots \mathrm{P}(c_K = c)$$
$$\cdot \mathbb{E}_{\bar{x}_m \sim \rho_{i_m}} \max_{a', a''} \|f(a''(\bar{x}_m)) - f(a'(\bar{x}_m))\|, \tag{40}$$

Note that

$$\mathbb{E}_c \sum_{k=0}^{K} \sum_{\{j_1, \ldots, j_K\} \in \mathrm{re}([K])} \sum_{i_1 \neq c} \cdots \sum_{i_k \neq c} \mathrm{P}(c_{j_1} = i_1) \ldots \mathrm{P}(c_{j_k} = i_k) \mathrm{P}(c_{j_{k+1}} = c) \ldots \mathrm{P}(c_K = c)$$
$$\cdot \max_{a', a''} \mathbb{E}_{\bar{x}_m \sim \rho_{i_m}} \|f(a''(\bar{x}_m)) - f(a'(\bar{x}_m))\|$$

$$= \mathbb{E}_c \sum_{k=0}^{K} \sum_{\{j_1, \ldots, j_K\} \in \mathrm{re}([K])} \sum_{i_m \neq c} \mathrm{P}(c_{j_m} = i_m) \cdot \max_{a', a''} \mathbb{E}_{\bar{x}_m \sim \rho_{i_m}} \|f(a''(\bar{x}_m)) - f(a'(\bar{x}_m))\|$$
$$\cdot \sum_{i_1 \neq c} \cdots \sum_{i_k \neq c} \mathrm{P}(c_{j_1} = i_1) \ldots \mathrm{P}(c_{j_k} = i_k) \mathrm{P}(c_{j_{k+1}} = c) \ldots \mathrm{P}(c_K = c)$$

$$= \mathbb{E}_c \mathbb{E}_{c' \neq c} \max_{a', a''} \mathbb{E}_{\bar{x}_m \sim \rho_{c'}} \|f(a''(\bar{x}_m)) - f(a'(\bar{x}_m))\|$$
$$\cdot \sum_{k=0}^{m} \sum_{\{j_1, \ldots, j_K\} \in \mathrm{re}([K])} \mathrm{P}(c_{j_1} \neq c) \ldots \mathrm{P}(c_{j_k} \neq c) \mathrm{P}(c_{j_{k+1}} = c) \ldots \mathrm{P}(c_K = c)$$
$$\leq \mathbb{E}_c \mathbb{E}_{c' \neq c} \max_{a', a''} \mathbb{E}_{\bar{x}_m \sim \rho_{c'}} \|f(a''(\bar{x}_m)) - f(a'(\bar{x}_m))\|$$
$$= \mathbb{E}_c \mathbb{E}_{c' \neq c} \mathbb{E}_{\bar{x}' \sim \rho_{c'}} \max_{a', a''} \|f(a''(\bar{x}')) - f(a'(\bar{x}'))\|. \tag{41}$$

Then by Corollary B.1, we have

$$\mathcal{R}^{\mathrm{un}}(f) \geq \bar{\mathcal{R}}^{\sup}(f) - \mathbb{E}_c \mathbb{E}_{\bar{x} \sim \rho_c} \mathbb{E}_a [\mathbb{E}_{\bar{x}' \sim \rho_c} \mathbb{E}_{a'} \min_{a''} \|f(a'(\bar{x})) - f(a''(\bar{x}'))\| + \mathbb{E}_{\bar{x}' \sim \rho_c} \max_{a', a''} \|f(a''(\bar{x}')) - f(a'(\bar{x}'))\|]$$
$$- \mathbb{E}_c \mathbb{E}_{\bar{x} \sim \rho_c} \mathbb{E}_a \mathrm{P}(c_m \neq c) \cdot [2\mathbb{E}_{\bar{x} \sim \rho_c} \mathbb{E}_a \|f(a(\bar{x})) - f(\bar{x})\| + \mathbb{E}_{\bar{x}' \sim \rho_c} \max_{a', a''} \|f(a''(\bar{x}')) - f(a'(\bar{x}'))\|]$$
$$- \mathbb{E}_c \mathbb{E}_{\bar{x} \sim \rho_c} \mathbb{E}_a \mathbb{E}_{c' \neq c} \mathbb{E}_{\bar{x}' \sim \rho_{c'}} \max_{a', a''} \|f(a''(\bar{x}')) - f(a'(\bar{x}'))\|$$
$$= \bar{\mathcal{R}}^{\sup}(f) - \mathbb{E}_c \mathbb{E}_{\bar{x} \sim \rho_c} \mathbb{E}_{\bar{x}' \sim \rho_c} \mathbb{E}_{a'} \min_{a''} \|f(a'(\bar{x})) - f(a''(\bar{x}'))\| - \mathbb{E}_c \mathbb{E}_{\bar{x} \sim \rho_c} \max_{a', a''} \|f(a''(\bar{x})) - f(a'(\bar{x}))\|$$
$$- 2\mathbb{E}_c \mathrm{P}(c' \neq c) \cdot \mathbb{E}_{\bar{x} \sim \rho_c} \mathbb{E}_a \|f(a(\bar{x})) - f(\bar{x})\| - \mathbb{E}_c \mathbb{E}_{\bar{x} \sim \rho_c} \max_{a', a''} \|f(a''(\bar{x})) - f(a'(\bar{x}))\|$$
$$- \mathbb{E}_{c'} \mathbb{E}_{c \neq c'} \mathbb{E}_{\bar{x}' \sim \rho_{c'}} \max_{a', a''} \|f(a''(\bar{x}')) - f(a'(\bar{x}'))\|$$
$$\geq \bar{\mathcal{R}}^{\sup}(f) - \mathbb{E}_c \mathbb{E}_{\bar{x} \sim \rho_c} \mathbb{E}_{\bar{x}' \sim \rho_c} \mathbb{E}_a \min_{a'} \|f(a(\bar{x})) - f(a'(\bar{x}'))\| - 5\mathbb{E}_c \mathbb{E}_{\bar{x} \sim \rho_c} \max_{a, a'} \|f(a(\bar{x})) - f(a'(\bar{x}))\|. \tag{42}$$

$\square$

*Proof of Theorem 2.1.* By combining Theorem 2.7 and Lemma 2.8, we finish the proof. $\square$

## B.3. Proof of the Improved Bound

**Theorem B.2.** *Under Assumption 2.2, we have*

$$r_k(i_1, \ldots, r_k) \geq \log \left(1 + \sum_{m=1}^{k} \exp \left(- \mathbb{E}_{\bar{x}' \sim \rho_c} \mathbb{E}_{\bar{x}_m \sim \rho_{i_m}} f(a(\bar{x}))^\top [f(\bar{x}') - f(\bar{x}_m)]\right) + (K - k)\right)$$

$$- [\mathbb{E}_{\bar{x}' \sim \rho_c} \mathbb{E}_{a'} \min_{a''} \|f(a'(\bar{x})) - f(a''(\bar{x}'))\| + \mathbb{E}_{\bar{x}' \sim \rho_c} \max_{a',a''} \|f(a''(\bar{x}')) - f(a'(\bar{x}'))\|]. \tag{43}$$

*Proof of Theorem B.2.* Under Assumption 2.2, we have

$$\mathbb{E}_{\bar{x}' \sim \rho_c} \mathbb{E}_{\bar{x}_m \sim \rho_{i_m}} \mathbb{E}_{a'',a_m} f(a(\bar{x}))^\top [f(a''(\bar{x}')) - f(a_m(\bar{x}_m))] = \mathbb{E}_{\bar{x}' \sim \rho_c} \mathbb{E}_{\bar{x}_m \sim \rho_{i_m}} f(a(\bar{x}))^\top [f(\bar{x}') - f(\bar{x}_m)]. \tag{44}$$

Then by (32), (34), and (35), we have

$$r_k(i_1, \ldots, r_k) \geq \log \Big( 1 + \sum_{k=1}^{k} \exp \big( - \mathbb{E}_{\bar{x}_m \sim \rho_{i_m}} \mathbb{E}_{a',a_m} f(a(\bar{x}))^\top [f(a'(\bar{x})) - f(a_m(\bar{x}_m))] \big)$$
$$+ \sum_{m=k+1}^{K} \exp \big( - \mathbb{E}_{\bar{x}_m \sim \rho_c} \mathbb{E}_{a',a_m} f(a(\bar{x}))^\top [f(a'(\bar{x})) - f(a_m(\bar{x}_m))] \big) \Big)$$

$$\geq \log \Big( 1 + \sum_{m=1}^{k} \exp \big( - [\mathbb{E}_{\bar{x}' \sim \rho_c} \mathbb{E}_{\bar{x}_m \sim \rho_{i_m}} f(a(\bar{x}))^\top [f(\bar{x}') - f(\bar{x}_m)]$$
$$+ \mathbb{E}_{\bar{x}' \sim \rho_c} \mathbb{E}_{a'} \min_{a''} \|f(a'(\bar{x})) - f(a''(\bar{x}'))\| + \mathbb{E}_{\bar{x}' \sim \rho_c} \max_{a',a''} \|f(a''(\bar{x}')) - f(a'(\bar{x}'))\|])$$
$$+ (K - k) \exp \big( - [\mathbb{E}_{\bar{x}' \sim \rho_c} \mathbb{E}_{a'} \min_{a''} \|f(a'(\bar{x})) - f(a''(\bar{x}'))\| + \mathbb{E}_{\bar{x}' \sim \rho_c} \max_{a',a''} \|f(a''(\bar{x}')) - f(a'(\bar{x}'))\|]) \Big)$$

$$= \log \Big( 1 + \Big[ \sum_{m=1}^{k} \exp \big( - \mathbb{E}_{\bar{x}' \sim \rho_c} \mathbb{E}_{\bar{x}_m \sim \rho_{i_m}} f(a(\bar{x}))^\top [f(\bar{x}') - f(\bar{x}_m)] \big) + (K - k) \Big]$$
$$\cdot \exp \big( - [\mathbb{E}_{\bar{x}' \sim \rho_c} \mathbb{E}_{a'} \min_{a''} \|f(a'(\bar{x})) - f(a''(\bar{x}'))\| + \mathbb{E}_{\bar{x}' \sim \rho_c} \max_{a',a''} \|f(a''(\bar{x}')) - f(a'(\bar{x}'))\|]) \Big)$$

$$\geq \log \Big( 1 + \sum_{m=1}^{k} \exp \big( - \mathbb{E}_{\bar{x}' \sim \rho_c} \mathbb{E}_{\bar{x}_m \sim \rho_{i_m}} f(a(\bar{x}))^\top [f(\bar{x}') - f(\bar{x}_m)] \big) + (K - k) \Big)$$
$$- [\mathbb{E}_{\bar{x}' \sim \rho_c} \mathbb{E}_{a'} \min_{a''} \|f(a'(\bar{x})) - f(a''(\bar{x}'))\| + \mathbb{E}_{\bar{x}' \sim \rho_c} \max_{a',a''} \|f(a''(\bar{x}')) - f(a'(\bar{x}'))\|]. \tag{45}$$

$\square$

*Proof of Theorem B.2.* Combining Theorem 2.5, Corollary B.1, and Theorem B.2, we finish the conclusion. $\square$

## B.4. Proof of the Generalization Bound

*Proof of Theorem 2.4.* By combining Theorem 2.1 and Theorem 4.6 in Lei et al. (2023) with $G_2 = 1$ for the logistic loss in (2), we get the assertion. $\square$

## C. Experimental Results of TinyImagenet

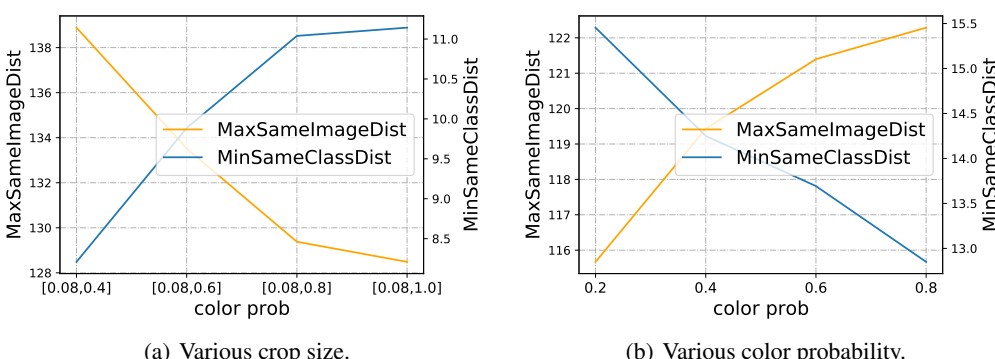

(a) Various crop size.

(b) Various color probability.

*Figure 7.* Pixel-level maximum distance between same-class different-image augmentations and minimum distance between same-image data augmentations on TinyImagenet.

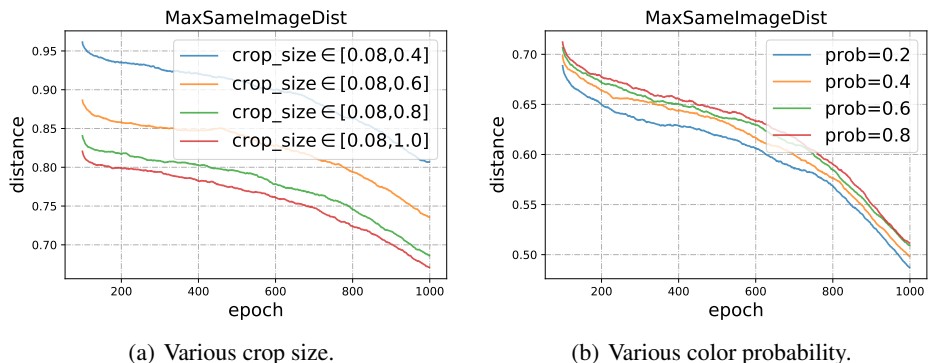

(a) Various crop size.

(b) Various color probability.

*Figure 8.* Representation-level maximum distance between same-class different-image augmentations on TinyImagenet.

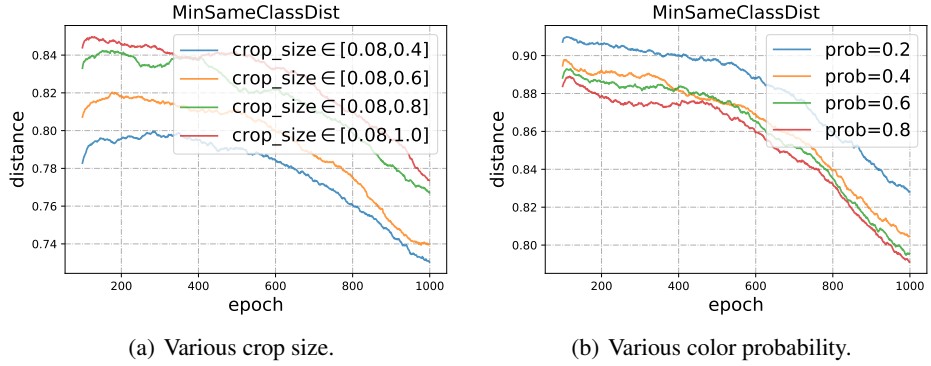

(a) Various crop size.

(b) Various color probability.

*Figure 9.* Representation-level minimum distance between different same-image data augmentations on TinyImagenet.

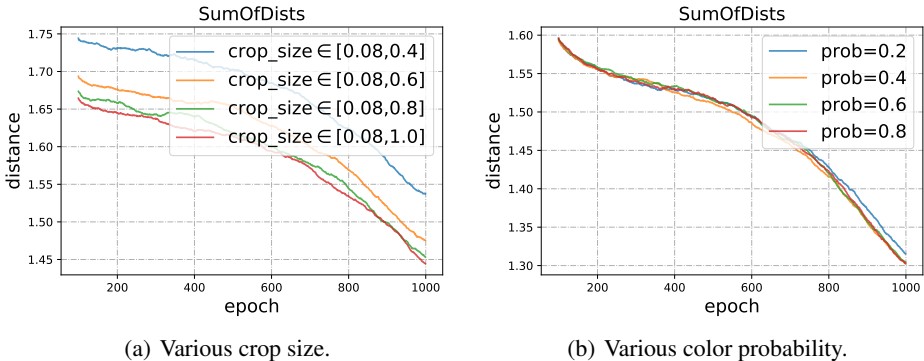

(a) Various crop size.

(b) Various color probability.

*Figure 10.* Sum of the two distance terms under various data augmentations in the embedding space on TinyImagenet.

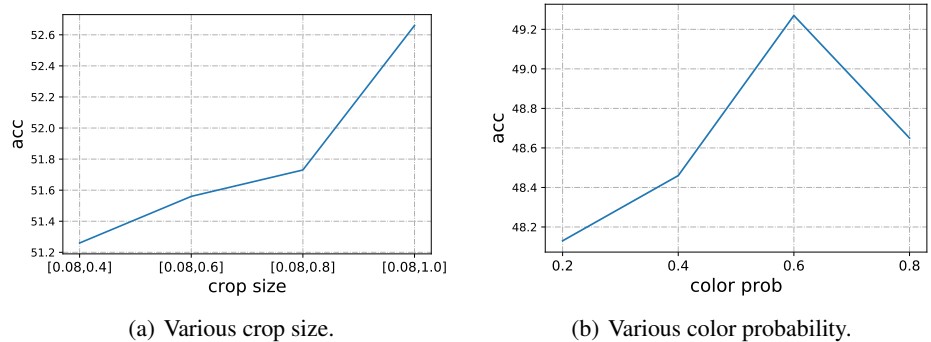

(a) Various crop size.

(b) Various color probability.

*Figure 11.* Linear probing accuracy under different data augmentation parameters on TinyImagenet.

