# OpenReview forum: "An Augmentation-Aware Theory for Self-Supervised Contrastive Learning"
_ICML.cc/2025/Conference — ICML 2025 poster_

### Official Review · Reviewer_Do5D · 2025-02-26

**Overall Recommendation:** 2

**Summary:**

This paper explores the role of data augmentation from the perspective of theoretical research. It for the first time propose an augmentation-aware error bound for self-supervised contrastive learning, showing that the supervised risk is bounded not only by the unsupervised risk, but also explicitly by a trade-off induced by data augmentation. It also discuss how certain augmentation methods affect the error bound. Some experiments verify its proposed theoretical results.

**Claims And Evidence:**

Yes.

**Essential References Not Discussed:**

No.

**Experimental Designs Or Analyses:**

No. Their experiments are intuitive.

**Methods And Evaluation Criteria:**

Yes.

**Other Comments Or Suggestions:**

Because of Question 6, I have not read the analysis of Section 3 carefully. If the authors provide some reasonable explanations for my concerns, I will improve my score.

**Other Strengths And Weaknesses:**

Strengths:

1.This paper for the first time proposes an augmentation-aware error bound for self-supervised contrastive learning, which is quite interesting.

Weaknesses:

1.In the lines 103-105, the symbol of the set of all possible data points should be $\bar{X}$ instead of $P_{\bar{X}}$.

2.In the second paragraph of Section 2.1, the first sentence should be “In unsupervised contrastive learning, we select the different data augmentations of the same input image as the positive samples, and select the data augmentations of different input images as negative samples”.

3.On the right head of the line 82, there lacks a period between “assumption”and “Moreover”.

4.The equation between the line 138 and 139 lacks a minus sign in the exponential function.

5.There are some mistakes in proof. For example, (1) the decomposition at the beginning of Proof of Theorem 2.7 should have $p_{K-1}(...) r_{K-1}(...)$; (2) the last equation of Equation (31) should be $\sum_{m=1}^k$.

6.The setting of Equation (26) is not very rigorous. Therefore, Equation (25) and Equation (29) are not strictly equivalent. Besides, it may be necessary to make some modifications to Theorem 2.6 and its proof.

7.On the right head of the line 309, “likewise” should be “Likewise”.

8.Equation (21) exceeds the page margin.

**Questions For Authors:**

1.In the third paragraph of Introduction, the authors state that the augmentation overlap assumption is still barely realistic, because in practice, it is unlikely for two different real images to share **exactly** the same augmented views. However, in my opinion, if an image has an augmented view having the same semantics as another image, we can say that these two images are augmentation overlapping.

2.What is the relationship between the distributions $P_X$ and $P_{\bar{X}}$?

3.Which results in Arora et al., 2019 and Nozawa &Sato, 2021 does Lemma 2.8 represent?

4.The authors state that they derive a novel decomposition of unsupervised contrastive risk. However, from my perspective, it is just a detailed expansion of unsupervised contrastive risk. What are the advantages of this decomposition?

5.What is the purpose that the authors provide the generalization bound in Theorem 2.4? It seems to be not essential. Please have more discussions about Theorem 2.4.

6.Assumption 3.1 is very confusing. Generally, Lipschitz continuity condition is $\|f(w) - f(w^\prime)\|\leq \|w-w^\prime\|$, where $w,w^\prime$ represent model parameters. Some works denote model parameter as $x$. However, in this paper, $x$ represents sample (image). Therefore, I suggest the authors carefully explain this assumption (it is better to give some examples) and check their subsequent analysis.

**Relation To Broader Scientific Literature:**

Please see the above Summary.

**Theoretical Claims:**

Yes. I have carefully check the correctness of all proofs in this paper. Some issues I found are listed behind.

---

> ### Author Rebuttal · Authors · 2025-04-01
>
> We express our sincere gratitude to Reviewer Do5D for appreciating the novelty of our theory. We address your concerns below.
>
> ---
>
> **Weaknesses.**
>
> **W1,W2,W3,W4,W7,W8.**
> **A.** We will revise the typos as suggested.
>
> **W5.** Mistakes in proof. (1) the decomposition at the beginning of Proof of Theorem 2.7 should have $p_{K-1}(\ldots)r_{K-1}(\ldots)$; (2) the last equation of Eq(31) should be $\sum_{m=1}^k$.
>
> **A.** **(1)** The typo $r_K$ in the second line should be $r_{K-1}$, so that $p_{K-1}r_{K-1}$ appears in the second term of the decomposition. This typo does not affect the overall correctness of the proof.
>
> **(2)** No, the summation should be $\sum_{m=k+1}^K$. According to the definition of $r_k$, there are $k$ different-class negatives, so the summation $\sum_{m=1}^K$ here breaks into two groups: the negatives $\bar{x}\_m$ having different classes from the anchor $\bar{x}$ ($\sum_{m=1}^k$) and the negatives $\bar{x}\_m$ sharing the same classes as the anchor $\bar{x}$ ($\sum\_{m=k+1}^K$). By fixing the abuse of notation $k$, Eq(31) becomes
> $$
> \log(1+\sum_{m=1}^k\exp(-\mathbb{E}\_{\bar{x}\_m\sim\rho_{i_m}}\mathbb{E}\_{a',a_m}f(a(\bar{x}))^\top[f(a'(\bar{x}))-f(a_m(\bar{x}\_m))])+\sum_{m=k+1}^K\exp(-\mathbb{E}\_{\bar{x}_m\sim\rho_c}\mathbb{E}\_{a',a_m}f(a(\bar{x}))^\top[f(a'(\bar{x}))-f(a_m(\bar{x}\_m))])),
> $$
> which simplifies the expectations and separates the two groups.
>
> **W6.** Eq(26) not rigorous, making Eq(25) and Eq(29) not strictly equivalent. Necessity to modify Theorem 2.6.
>
> **A.** **(1)** Eq(26) is rigorous according to the commutative law of addition. It holds for any $\\{i_1,\ldots,i_K\\}$ being a rearrangement of $[K]$. Eq(26) ensures that e.g. $\mathbb{E}\_{\bar{x}_1\sim\rho_1}\mathbb{E}\_{\bar{x}_2\sim\rho_2}\mathcal{L}=\mathbb{E}\_{\bar{x}_1\sim\rho_2}\mathbb{E}\_{\bar{x}_2\sim\rho_1}\mathcal{L}$, and therfore Eq(27) can all be represented by the same $r_k$, making Eq(25) and Eq(29) strictly equivalent.
>
> **(2)** Theorem 2.6 is correct. Typos will be fixed according to **W5(2)**.
>
> ---
>
> **Questions.**
>
> **Q1.** Understanding of augmentation overlapping.
>
> **A1.** We respectively disagree from a mathematical viewpoint. In the original paper (Wang et al., 2021), augmentation overlap is formulated as $\mathcal{T}$-connectivity (Definition 4.3), where $x_i$ and $x_j$ are $\mathcal{T}$-connected if $\mathrm{supp}(p(x_i^+|x_i))\cap\mathrm{supp}(p(x_j^+|x_j))\neq\emptyset$, or equivalently, $\exists t_i,t_j \in \mathcal{T}$ s.t. $t_i(x_i)=t_j(x_j)$, meaning that $x_i$ and $x_j$ sharing exactly the same augmented views $t_i(x_i)=t_j(x_j)$, which is unrealistic.
>
> ---
>
> **Q2.** Relationship between $P_X$ and $P_{\bar{X}}$.
>
> **A2.** $\mathrm{P}_X$ is the distribution of $a(\bar{x})$, where $a\sim\mathrm{P}_A$ and $\bar{x}\sim\mathrm{P}\_{\bar{X}}$.
>
> ---
>
> **Q3.** Which results does Lemma 2.8 represent?
>
> **A3.** Lemma 2.8 is part of Proposition 6 in Nozawa & Sato, 2021 and Lemma 4.3 in Arora et al., 2019. Compared with Proposition 6 in Nozawa & Sato, 2021, Lemma 2.8 does not have the $d(f)$ term and shows an equation because the left hand side is $\bar{\mathcal{R}}^{\mathrm{sup}}$ instead of $\bar{\mathcal{R}}^{\mathrm{un}}$ (without the need of using Jensen's inequality).
>
> ---
>
> **Q4.** Advantages of decomposition.
>
> **A4.** The decomposition makes separation between same-class and different-class negative terms possible. Specifically, it decomposes the contrastive risk according to the number of negative samples having different labels from the anchor sample. The decomposition is non-trivial, as it only works for contrastive losses that treat negative samples equally (Eq(26)). With this decomposition, we further decompose the risk into different-class negative terms and same-class negatives terms (Eq(31)). Then respectively investigating the two types of terms (Eq(33) and Eq(37)), we reach our main theorems.
>
> ---
>
> **Q5.** Discussions about Theorem 2.4.
>
> **A5.** Theorem 2.4 shows the finite-sample error bound as a complement to the other theorems based on the population distribution. It also shows the compatibility of our bound with generalization bounds in previous works.
>
> ---
>
> **Q6.** Explain Lipschitz continuity.
>
> **A6.** The Lipschitz continuity condition concerning $f$ with respect to the image $x$ for convolution neural networks has been widely made in previous works, see e.g.
> [1] On Lipschitz Bounds of General Convolutional Neural Networks. TIP, 2019.
> [2] Intriguing properties of neural networks. ICLR, 2014.
> Moreover, the consistency between our pixel-level (Figure 2) and representation-level (Figures 3 & 4) experiments also verifies the assumption.
>
> ---
>
> Thanks for your insightful and constructive comments. According to our answers, we have clarified that our proofs have no factual mistakes (only typos to be fixed). We sincerely suggest the reviewer continue reviewing Section 3 and reconsider the overall recommendation score. We welcome any further questions during your follow up reviews.

---

> > ### Comment · Reviewer_Do5D · 2025-04-02
> >
> > Thanks for the authors' response. Firstly, I want to explain **W5 (2)**. There is a misunderstanding that $\sum_{k=1}^k \rightarrow \sum_{m=1}^k$ rather than $\sum_{m=k+1}^K \rightarrow \sum_{m=1}^k$. Besides, I still have some questions.
> >
> > **1.** For **W6**, Eq (28) not Eq (26) is not very rigorous. For example, $c_{j_k}=i_k$ and $c_{j_{k+1}=\cdots=c_K=c}$ in Eq (28). While $c_j=i_j$ in Eq (25). Therefore, the symbol settings are a bit messy. So, it may be necessary to make some modifications to Theorem 2.5 (not Theorem 2.6) and its proof.
> >
> > **2.** The author's requirement for augmentation overlap is too strict. I really can't accept this viewpoint. Taking Fig 1 (b) in Wang et al., 2021 as an example, do the authors think that the two augmented views related to tires are not-overlap? If so, the two areas of tires in Fig 1 of this paper have different semantics. It is contradictory.
> >
> > **3.** For **Q3**, I still don't understand how $\bar{\mathcal{R}}^{un}$  became $\bar{\mathcal{R}}^{sup}$. And, I didn't find $\bar{\mathcal{R}}^{un}$. Could the authors provide a detailed demonstration?

---

> > > ### Author Response · Authors · 2025-04-02
> > >
> > > We thank the reviewer's timely reply. We address your concerns as follows.
> > >
> > > ---
> > > **Q1.** For **W6**, Eq (28) not Eq (26) is not very rigorous. For example, $c_{j_k}=i_k$ and $c_{j_{k+1}}=\ldots=c_K=c$ in Eq(28).  While $c_j=i_j$ in Eq (25). Therefore, the symbol settings are a bit messy. So, it may be necessary to make some modifications to Theorem 2.5 (not Theorem 2.6) and its proof.
> > >
> > > **A1.** Eq (28) is a rigorous definition. The condition on the right hand side means that there exist $(K-k)$ negative samples sharing the same class $c$, and the other $k$ negative samples having classes $\{i_1,\ldots,i_k\}$. The notation $p_k(i_1,\ldots,i_k)$ depends only on $\{i_1,\ldots,i_k\}$, and does not depend on any specific $\{j_1,\ldots,j_k\}$. Besides, Eq (25) rigorously equals to Eq (29). For example, let us examine $k=1$ (the second last line in Eq (25)),
> > > $$
> > > \mathbb{E}_c \mathbb{E}\_{\bar{x}\sim\rho_c}\mathbb{E}_a\sum\_{i_j\neq c} \sum\_{j=1}^K \mathrm{P}(c_1=c)\cdots\mathrm{P}(c_j=i_j)\cdots\mathrm{P}(c_K=c) \mathbb{E}\_{\bar{x}_k\sim\rho_c,k\neq j} \mathbb{E}\_{\bar{x}_j\sim\rho\_{i_j}} \mathbb{E}\_{a',\\{a_k\\}\_{k\in[K]}} \mathcal{L}(a(\bar{x}),a'(\bar{x}),a_k(\bar{x}_k);f)
> > > $$
> > > $$
> > > =\mathbb{E}_c\mathbb{E}\_{\bar{x}\sim\rho_c}\mathbb{E}\_a \sum\_{i_j\neq c}\sum\_{j=1}^K\mathrm{P}(c_1=c) \cdots\mathrm{P}(c_j=i_j)\cdots\mathrm{P}(c_K=c) r_1(i_j)
> > > $$
> > > $$
> > > =\mathbb{E}_c\mathbb{E}\_{\bar{x}\sim\rho_c}\mathbb{E}\_a \sum\_{i_j\neq c} p_1(i_j)r_1(i_j),
> > > $$
> > >
> > > where the first equation holds because of Eq (26) and Eq (27), and the second equation holds because by Eq (28)
> > > $$
> > > p_k(i_j)= \mathrm{P}(\exists j, \text{ such that } c_j=i_j \text{ and } c_\ell=c \text{ for } \ell\neq j)
> > > $$
> > > $$
> > > =\mathrm{P}(c_1=i_j)\mathrm{P}(c_2=c)\cdots\mathrm{P}(c_K=c) + \mathrm{P}(c_1=c)\mathrm{P}(c_2=i_j)\cdots\mathrm{P}(c_K=c)
> > > +\ldots + \mathrm{P}(c_1=c)\mathrm{P}(c_2=c)\cdots\mathrm{P}(c_K=i_j)
> > > $$
> > > $$
> > > := \sum_{j=1}^K \mathrm{P}(c_1=c)\cdots\mathrm{P}(c_j=i_j)\cdots\mathrm{P}(c_K=c).
> > > $$
> > > The calculations of other lines in Eq (25) are similar, only with a bit more complicated notations required.
> > >
> > > ---
> > > **Q2.** The author's requirement for augmentation overlap is too strict. I really can't accept this viewpoint. Taking Fig 1 (b) in Wang et al., 2021 as an example, do the authors think that the two augmented views related to tires are not-overlap? If so, the two areas of tires in Fig 1 of this paper have different semantics. It is contradictory.
> > >
> > > **A2.** We would like to clarify more on the understanding of augmentation overlap.
> > >
> > > Firstly, in the 4th paragraph of the introduction in Wang et al., 2021, the two views in Figure 1(b) are commented as "very much alike that we could even hardly tell them apart", and *augmentation overlap* is defined as "there will be support overlap between different intra-class images through aggressively augmented views of them". The term "support overlap" indicates the existence of the same augmented views between different intra-class images.
> > >
> > > Secondly, if two views are similar but not exactly the same, they can still have the same semantics, e.g. two similar but not exactly the same wheels taken from different car figures share the same semantics of "wheel". (Perhaps the reviewer has different understandings of "semantics". Here, by "semantics", we refer to the semantic labels illustrated in Section 3.1 of our submission.)
> > >
> > > Thirdly, if augmentation overlap is interpreted as two images having the same semantics, one cannot mathematically explain how the model recognizes the semantics without making additional assumptions.
> > >
> > > Nonetheless, to avoid possible confusion, we will tone down our remark about augmentation overlap, and remove the statement from our next version.
> > >
> > > ---
> > > **Q3.** For **Q3**, I still don't understand how $\bar{\mathcal{R}}\_{\mathrm{un}}$ became $\bar{\mathcal{R}}\_{\mathrm{sup}}$. And, I didn't find $\bar{\mathcal{R}}_{\mathrm{un}}$. Could the authors provide a detailed demonstration?
> > >
> > > **A3.** $\bar{\mathcal{R}}_{\mathrm{sup}}$ defined in our paper equals the first term of Eq (7) in Nozawa & Sato, 2021, and also Inequality (b) in the proof of Lemma 4.3 in Arora et al., 2019. According to the proof of Proposition 6 in Nozawa & Sato, 2021 and the proof of Lemma 4.3 in Arora et al., 2019, it further decomposes into $(1-\tau_K)\cdots +\tau_K\cdots$.
> > >
> > > In previous works, the relationship between $\mathcal{R}\_{\mathrm{un}}$ and $\bar{\mathcal{R}}\_{\mathrm{sup}}$ is built via Jensen's inequality (Eq (7) in Nozawa & Sato, 2021, and Inequality (b) in the proof of Lemma 4.3 in Arora et al., 2019). In our submission, it is presented in Theorem 2.7. The typo $\bar{\mathcal{R}}\_{\mathrm{un}}$ should be $\mathcal{R}\_{\mathrm{un}}$ (defined in Eq (3)) in our last reply.
> > >
> > > ---
> > > Thanks again for the timely reply. We hope our response solves all your additional concerns, as this is the last rebuttal we can post. We strongly suggest the reviewer finish reviewing the rest of Section 3 and reconsider the recommendation score.

---

### Official Review · Reviewer_6JLt · 2025-02-28

**Overall Recommendation:** 3

**Summary:**

In this paper, the authors theoretically study how augmentations affect supervised risk, an unexplored area in self-supervised learning. Furthermore, they conduct some experiments to verify the theories.

**Claims And Evidence:**

Yes, the claims are supported by evidence.

**Essential References Not Discussed:**

None

**Experimental Designs Or Analyses:**

After reviewing the experiments, I think the findings are reliable.

**Methods And Evaluation Criteria:**

There is not a new method in this paper.

**Other Comments Or Suggestions:**

None

**Other Strengths And Weaknesses:**

Strengths
1. This paper supports the augmentation study and highlights its important role in self-supervised learning.

Weakness
1. The error bound is not directly correlated with the final performance.
2. The paper lacks empirical guidance, making it unhelpful for selecting appropriate augmentation parameters.

**Questions For Authors:**

1. When selecting augmentation parameters, only suitable values lead to good performance. Could the authors explain why this occurs based on these theorems?

**Relation To Broader Scientific Literature:**

None

**Theoretical Claims:**

No, I am not familiar with it.

---

> ### Author Rebuttal · Authors · 2025-04-01
>
> We express our sincere gratitude to Reviewer 6JLt for appreciating the significance and reliability of our findings. We address your concerns below.
>
> ---
>
> **Q1.** The error bound is not directly correlated with the final performance.
>
> **A1.** We respectfully disagree. As shown in Section 4.2, we show that the optimal data augmentation parameter minimizing the distance sum also leads to the best downstream accuracy, indicating that lower downstream error bounds highly correlate with better downstream performance. Specifically, in Theorem 2.3 we show that the supervised error bound depends not only on the unsupervised loss but also on the maximum distance between augmentations and the minimum distance between samples of the same class. Typically, efforts focus on minimizing the unsupervised error to reduce the supervised error bound. However, the two distance terms, which depend on the augmentation parameter, are also crucial and must be minimized to improve classification accuracy. This is empirically verified by Figures 5 and 6, which show that classification accuracy is inversely proportional to the sum of these two distance terms.
>
> ---
>
> **Q2.** The paper lacks empirical guidance, making it unhelpful for selecting appropriate augmentation parameters.
>
> **A2.** On the one hand, our theoretical results in Section 2.2 show that the downstream classification risk depends not only on the unsupervised loss but also on the sum of two distance terms related to data augmentation, which provides empirical insight that the selection of data augmentation parameters is also vitally important. On the other hand, Section 3 demonstrates a theoretical trade-off between the two distance terms, indicating the existance of the optimal data augmentation parameters. Empirical varifications show that the optimal range of crop size should be large enough, with the upper bound being 1.0; the optimal probability of color jitter is around 0.6-0.8. (See Figures 5 & 6 and also the additional experiments on Imagenet100 in the response **A2** to Reviewer Xxrg.) Furthermore, for future works, our theoretical results also guides a possible direction for designing new data augmentation methods to deliberately minimize some unsupervised surrogate of the two distances.
>
> ---
>
> **Q3.** When selecting augmentation parameters, only suitable values lead to good performance. Could the authors explain why this occurs based on these theorems?
>
> **A3.** In Theorems 2.1 and 2.3, we show that the supervised error bound depends not only on the unsupervised loss but also on the maximum distance between augmentations and the minimum distance between samples of the same class. Through model training, the unsupervised error can be reduced to a very small value. However, the impact of these two distance terms on classification performance is often overlooked. In fact, these distance terms are highly influenced by the augmentation parameter values. The augmentation parameter values that minimize the sum of these two distances may also yield the smallest classification error bound, as verified by Figures 5 and 6.
>
> ---
>
> Thanks for your insightful and constructive comments. Hope our explanations and additional experiments can address your concerns.

---

> > ### Comment · Reviewer_6JLt · 2025-04-03
> >
> > Thank you for addressing my concerns. I will maintain my original score.

---

### Official Review · Reviewer_cGdD · 2025-03-10

**Overall Recommendation:** 3

**Summary:**

Self-supervised contrastive learning effectively extracts representations from unlabeled data. Despite its success prompting theoretical studies, the impact of specific data augmentation techniques is still under-explored. To address this, the authors proposed an augmentation-aware error bound for self-supervised contrastive learning, showing that the supervised risk is bounded not only by the unsupervised risk, but also explicitly by a trade-off induced by data augmentation. Furthermore, the authors discussed how certain augmentation methods affect the error bound.

## Update after rebuttal

After reviewing the author's rebuttal and the reviews from another reviewer, I will keep my current weak accept rating.

**Claims And Evidence:**

The authors aim to formulate the problem and motivation using different scenarios:
1. Directly establishing a relationship between unsupervised contrastive risk and supervised risks through statistical modeling.
2. Relying on the assumption of an augmentation graph and borrowing mathematical tools from unsupervised spectral clustering.
3. Exploring other explanatory works of contrastive learning from the perspective of feature geometry.

With these three scenarios, the authors attempt to formulate the problem, which remains under-explored. Thus, they propose an augmentation-aware theory for self-supervised contrastive learning, decomposing the unsupervised contrastive risk in relation to the number of negative samples sharing the same label as the anchor. Furthermore, they analyze specific types of data augmentation and discuss the existence of a trade-off between the two distance terms concerning the strength of data augmentation.

**Essential References Not Discussed:**

Some of the recent papers are discussed in the theoretical sections, but these discussed papers need to be addressed in the experimental section as well. It would be beneficial to compare their performance with the proposed approach and analyze how they affect performance in downstream tasks.

**Experimental Designs Or Analyses:**

The experimental design needs reorganization and more discussion based on the theoretical formulation (augmentation-aware error bound, impact of different augmentation, some downstream tasks). As noted, some experimental settings lack sufficient discussion and analysis in relation to the theoretical framework. The experimental results should be presented with different augmentation strategies based on their formulations, the Augmentation-Aware Error Bound compared to other approaches (as discussed in Section 2.3). These aspects should be addressed more thoroughly, as they represent a significant weakness of the paper.

**Methods And Evaluation Criteria:**

In Section Two, the authors introduce a mathematical formulation for unsupervised contrastive learning and downstream classification, accompanied by a detailed analysis of the proposed approach. Following this, an augmentation-aware error bound is presented, supported by intriguing mathematical theory. This section offers a compelling theoretical explanation of how the authors address the issue with clear formulations. However, the organization of Section Two is lacking. For instance, in the middle of the theoretical explanation, the authors introduce a discussion that feels disconnected from the main theoretical idea.

The authors design two types of experimental settings: pixel-level and representation-level, to validate the proposed approach. However, there is a lack of discussion regarding the contributions of the proposed method(in tense of ablation, comparison). In Section 2, the authors explain the main theoretical concepts with a strong promise of performance improvement. However, I cannot find a thorough discussion in these two experiment settings with different scenarios (impact of augmentation, augmentation-aware error bound). Furthermore, it is necessary to evaluate the proposed approach against state-of-the-art papers and to highlight its main contributions across different settings, as mentioned by the authors.

**Other Comments Or Suggestions:**

The authors need to check the entire paper and proofread it thoroughly. Some sentences without proper punctuation. For example: "That means, we no longer need the conditional independence assumption Moreover, we formulate the data..."

**Other Strengths And Weaknesses:**

I found the theoretical aspects of the paper is good. The problem formulation, arguments, and claims about how the approach is addressed in their work are interesting. Furthermore, their mathematical formulation is somehow fine.

However, there are some weaknesses in the paper, particularly in the organization of Section 2 and the lack of detailed discussion in the experimental section. Specifically, the experimental section needs a more thorough discussion, including comparisons with different variations and settings, as well as a comparison of the proposed approach with existing works related to their approach.

**Questions For Authors:**

Most of the comments and questions have been mentioned in the sections above. However, these issues still need to be addressed:

1. Do the authors analyze and discuss the impact of different augmentations and augmentation-aware error bounds in the two experimental settings?
2.	How does the proposed approach compare against state-of-the-art papers, and can the authors highlight its main contributions across these different settings?
3.	Since the paper serves as a foundation for further theoretical exploration of data augmentation (self-supervised and contrastive learning), it needs a detailed analysis of the experiments and discussions.

**Relation To Broader Scientific Literature:**

Their formulation is very interesting, particularly in relation to contrastive and self-supervised learning, as it addresses the risks associated with different augmentation techniques. Furthermore, the approach is intriguing for utilizing unlabeled samples in downstream classification tasks.

**Theoretical Claims:**

The authors provide a solid theoretical formulation of the proposed approach, clearly outlining how they intend to address the issues with precise mathematical formulations. However, the organization of the paper still needs some modifications. In certain sections, the authors introduce theoretical concepts alongside mathematical formulations, while simultaneously discussing experimental performance (Section 2.3). This structure can be confusing and may benefit from clearer separation.

---

> ### Author Rebuttal · Authors · 2025-04-01
>
> We express our sincere gratitude to Reviewer cGdD for appreciating the soundness of our theoretical formulations and results. We address your concerns below.
>
> ---
>
> **Q1.** Organization of Section 2.
>
> **A1.** As suggested, we will make the discussions in Section 2.3 an independent section following Section 2 in our next version and also add a roadmap in the introductory section to enhance readability.
>
> ---
>
> **Q2.** Lack of discussion regarding the contributions and comparisons with SOTA.
>
> **A2.** The discussions of our contribution locate mainly in Section 2.3, where we compare our bound with previous theoretical works. Besides, we also conduct verification experiments in Section 4 to verify our new theoretical claims proposed in Sections 3.2 and 3.3. As the main contribution of this paper is to theoretically explain the role of data augmentation in self-supervised contrastive learning, the significance of the proposed error bound is not necessarily verified by ablations or experimental comparisons. Nonetheless, we could compare empirically the bound value with previous theoretical works. In detail, as shown in Figures 5 and 10, the sum of the two distance terms is around 1 (1.3 for TinyImagenet and 0.8 for CIFAR100) after 1000 epoch training, whereas we additionally measure the contrastive loss value to be around 42 (41.6 for TinyImagenet and 41.2 for CIFAR100). This indicates that compared with previous bounds, our bound relax the data generation assumption with the bound value increases less than 3% (1/42). See also **A6**.
>
> ---
>
> **Q3.** Some experimental settings lack sufficient discussion and analysis in relation to the theoretical framework.
>
> **A3.** The experimental section 4.1 is specifically in relation to theoretical analysis in Section 3 verifying the trade-off between the two distance terms, and the experimental section 4.2 is in relation to the error bound of Theorem 2.3, where the optimal augmentation parameter leads to the minimum distance sum and also the highest downstream accuracy.
>
> To further enhance the relation of experiments to the theoretical framework, we here present an additional experiment about the formulations of data augmentation strategies. This experiment verifies the theoretical analysis in Section 3.3 about the combination of random crop and color distortion. Specifically, we present the pixel-level distances between data augmentations using only random crops and that using random crops plus color jitter on CIFAR100. As shown in the figure https://imgur.com/a/aj3Xp4i, on top of random crop, color distortion further reduces the minimum same-class different-image distance and enhances the maximum same-image different-augmentation distance.
>
> ---
>
> **Q4.** Missing punctuation.
>
> **A4.** We will fix the typo in our next version.
>
> ---
>
> **Q5.** Do the authors analyze and discuss the impact of different augmentations and augmentation-aware error bounds in the two experimental settings?
>
> **A5.** Yes, in both settings, we show that stronger augmentations, e.g. smaller range of crop size and higher probability of color jitter, lead to larger maximum same-image different-augmentation distance and smaller same-class different-image distance, which coincides with the theoretical analysis in Sections 3.2 and 3.3. From the perspective of our error bound, there exists a trade-off between the two distance terms w.r.t. the strength of data augmentation, indicating the existance of the optimal augmentation parameters. Related discussions can be found in Section 4.1.1 2nd paragraph, Section 4.1.2 2nd paragraph, and Section 4.2 2nd paragraph.
>
> ---
>
> **Q6.** How does the proposed approach compare against state-of-the-art papers, and can the authors highlight its main contributions across these different settings?
>
> **A6.** In fact, instead of proposing a new contrastive learning methods that competes against the state-of-the-art,, the aim of our error bound is to explain their working mechanisms. Our bound applies for any contrastive learnin methods as long as they use the InfoNCE loss function regardless of the backbones or other architectural techniques. It builds upon a more realistic data generation process described in Section 2.1 without relying on any further assumptions (Theorem 2.1). That is, compared with other theoretical papers deriving the error bound, our bound can easily adapt these previous bounds under our more realistic data generation, only with two additional trade-off distance terms added to these bounds. See also the third paragraph of Section 2.3 for more details.
>
> ---
>
> **Q7.** The paper needs a detailed analysis of the experiments and discussions.
>
> **A7.** See **A2** and **A3**.
>
> ---
>
> Thanks for your insightful and constructive comments. Hope our explanations and additional experiments can address your concerns.

---

### Official Review · Reviewer_Xxrg · 2025-03-14

**Overall Recommendation:** 3

**Summary:**

This paper theoretically examines the role of data augmentation in contrastive learning. It demonstrates that supervised risk is bounded not only by unsupervised risk but also by a trade-off introduced by data augmentation. The analysis is further extended using Lipschitz continuity, providing insights into how changes in input data or augmentations affect the model's output and error bounds. Experimental results validate the theoretical findings, reinforcing the impact of augmentation on contrastive learning performance.

**Claims And Evidence:**

Claims made in the submission are supported by clear evidence.

**Essential References Not Discussed:**

Most papers are discussed.

**Experimental Designs Or Analyses:**

The experiments are sound for evaluating the theorems introduced.

**Methods And Evaluation Criteria:**

There are no new methods proposed. Evaluation criteria make sense for the problem of the theoretical analysis.

**Other Comments Or Suggestions:**

Page 1, top right: The statement “we usually use only two views in training instead of using multiple ones.” may be misleading for readers unfamiliar with  SSL. While many traditional contrastive learning methods use two views, several recent approaches, such as MMCR and DINO, leverage multiple views.

**Other Strengths And Weaknesses:**

Strength:
1. This work extends previous theoretical analyses by making assumptions that may not fully align with real-world scenarios. This work does not assume that anchor and positive samples are conditionally independent or that two different real images share exactly the same augmented views. These assumptions, while useful for theoretical exploration, may not always hold in practical applications.
2. The experimental results validate the new theoretical results.

Weakness:
1. The experiments are conducted only on small-scale datasets such as CIFAR-10/100. Since image resolution and dataset complexity could influence the theoretical results, an evaluation of larger, high-resolution datasets would provide a more comprehensive validation.
2. The study focuses exclusively on InfoNCE loss, which may limit the generalization of the theoretical findings.

**Questions For Authors:**

Please address the questions regarding the assumptions and evaluations, as these concerns could impact my assessment of the paper.

**Relation To Broader Scientific Literature:**

Contrastive learning is a key self-supervised learning approach with numerous real-world applications. Augmentation plays an integral role in contrastive learning, often determining the quality of learned representations. Gaining a theoretical understanding of augmentation's impact could have significant implications for the broader scientific literature, shaping future advancements in self-supervised learning and representation learning.

**Theoretical Claims:**

I believe the analysis in Sections 2 and 3 is correct. However, Assumption 2.2 could benefit from some empirical verification. The radius $R$ in Theorem 2.4 seems to be not formally introduced and explained.

---

> ### Author Rebuttal · Authors · 2025-04-01
>
> We express our sincere gratitude to Reviewer Xxrg for appreciating our practical theoretical assumptions and novel theoretical results. We address your concerns below.
>
> ---
>
> **Q1.** Empirical verification of Assumption 2.2.
>
> **A1.** For each input image in the CIFAR100 dataset, we generate 100 different random augmentations, and plot the tsne visualizations of the random augmentated views, the mean of the augmented views, and the original input sample in the embedding space. We show the tsne plots of 4 randomly selected samples in https://imgur.com/a/NadpBSl. We see that the mean of augmented views lie near the embedding of the original input sample, which verifies Assumption 2.2. Besides, we also measure the $l$-2 distance between the mean of augmentations and the original sample in the embedding space. We show that on CIFAR100, the average distance is 0.083, which is small enough considering that the embeddings are normalized to have unit l-2 norm, indicating that the mean of augmented embeddings $\mathbb{E}_a f(a(\bar{x}))$ is close enough to input embedding $f(\bar{x})$.
>
> ---
>
> **Q2.** The experiments are conducted only on small-scale datasets such as CIFAR-10/100. Since image resolution and dataset complexity could influence the theoretical results, an evaluation of larger, high-resolution datasets would provide a more comprehensive validation.
>
> **A2.** Following your advice, we conduct additional verification experiments on Imagenet100, which contains 224*224 high resolution images from 100 classes. Due to time limitation, we run experiments with ResNet-18 with batchsize 64 for 50 epochs on 2 24G RTX 3090 GPUs. (The other settings follow those of TinyImagenet in our submitted paper.) The pixel-level verifications are shown in https://imgur.com/a/UcVPXwS, and the representation-level verification w.r.t. crop size is shown in https://imgur.com/a/jl00TVn, where the results conincides with that in Sections 4.1.1 and 4.1.2, i.e., as the augmentation strength increases, the max same-image different-augmentation distance increases and the min same-class different-image distance decreases. Besides, in the figure https://imgur.com/a/wPs9WOO, we verify that the optimal augmentation parameters with the smallest distance sum also leads to the highest downstream accuracy, which coincides with Section 4.2.
>
> ---
>
> **Q3.** The study focuses exclusively on InfoNCE loss, which may limit the generalization of the theoretical findings.
>
> **A3.** On the one hand, as one of the mainstream contrastive losses, InfoNCE-based contrastive learning still achieves state-of-the-art performances e.g. SimCLR, MoCo, etc. The extensive applications of InfoNCE loss partly demonstrate the generalization of our theory. On the other hand, although our theoretical results focus on the InfoNCE loss, we strongly believe that our analysis can potentially be generalized to other contrastive losses, e.g. BYOL, BarlowTwins, etc. Specifically, our analysis theoretically explains the effect of data augmentation to contrastive learning based only on the data generation procedure described in Section 2.1, with mild (Theorem 2.3) or even no further assumptions required (Theorem 2.1) and regardless of the network architectures. As long as a contrastive learning method selects positive/negative samples using random data augmentations, we can adapt the two trade-off distance terms to any error bound revealing the relationship between contrastive and downstream risks.
>
> ---
>
> **Q4.** Page 1, top right. The statement “we usually use only two views in training instead of using multiple ones.” may be misleading for readers unfamiliar with SSL. While many traditional contrastive learning methods use two views, several recent approaches, such as MMCR and DINO, leverage multiple views.
>
> **A4.** We will replace this statement with "we usually use only two views in training instead of using multiple ones in popular methods such as SimCLR and MoCo".
>
> ---
>
> Thanks for your insightful and constructive comments. We sincerely hope you can reconsider the recommendation score if our explanations and additional experiments solves your concerns. We also welcome further questions and discussions.

---

> > ### Comment · Reviewer_Xxrg · 2025-04-04
> >
> > Thank you for addressing my concerns. After carefully reviewing the authors’ responses and the discussions from other reviewers, I am happy to revise my original recommendation from a 2 to a 3. That said, I continue to share the concerns raised by Reviewer 6JLt regarding Questions 2 and 3.

---

> > > ### Author Response · Authors · 2025-04-07
> > >
> > > Thank you very much for your timely reply, and for raising the recommendation score. To address your further concerns, aside from the original reply to Q2&Q3 of Reviewer 6JLt, we provide additional elaborations as follows.
> > >
> > > ---
> > >
> > > **Q2 of Reviewer 6JLt.** The paper lacks empirical guidance, making it unhelpful for selecting appropriate augmentation parameters.
> > >
> > > **A.** The theoretical results in Section 2.2 and Section 3 provide empirical guidance that the downstream performance depends on the trade-off between the two distance terms related to data augmentation. For augmentations such as random crop and color distortion, the two distance terms are specifically related to the augmentation strength.
> > >
> > > To theoretically guide the selection of augmentation parameters, we refer to Theorem 3.2, indicating that the upper bound of supervised risk is related to the sum of the pixel-level distances. As suggested by Theorem 3.2, the augmentation parameters having the minimum pixel-level distance sum lead to the best downstream performance. We conduct additional validation of this empirical guidance by comparing the pixel-level sum of distances and the downstream linear probing accuracy. In Figure https://imgur.com/a/b99SQBr, we show that on all three benchmark datasets, the smaller pixel-level distance sum leads to better downstream accuracy. In other words, Theorem 3.2 provides the theoretical guidance that we can roughly select appropriate augmentation parameters by comparing the pixel-level distance sum before training.
> > >
> > > ---
> > >
> > > **Q3 of Reviewer 6JLt.** When selecting augmentation parameters, only suitable values lead to good performance. Could the authors explain why this occurs based on these theorems?
> > >
> > > **A.** "Only suitable values lead to good performance" because inappropriate parameters lead to a large distance sum, and consequently result in a worse downstream error bound (according to Theorems 2.1 and 2.3). For example, on Imagenet100, the worst-performed crop size parameter [0.08,0.4] has a representation-level distance sum of 1.69, whereas the distance sum of the best-performed parameter [0.08,1.0] is only 1.53. The larger distance sum leads to a worse downstream error bound and accordingly a worse downstream accuracy. The results on other datasets are similar, as evidenced by Figures 5, 6, 10, and 11.
> > >
> > > ---
> > >
> > > Thanks again for the reply. We hope our response solves all your additional concerns.

---

### Decision · Program_Chairs · 2025-05-01

**Decision:**

Accept (poster)

**Comment:**

This paper received ratings of 3,3,3,2. It offers a valuable theoretical contribution by introducing an augmentation-aware error bound for self-supervised contrastive learning, providing new insights into how data augmentation affects supervised risk. The analysis is rigorous and avoids overly restrictive assumptions, and while the experiments support the theory, they are limited to small-scale datasets and lack comprehensive comparisons. Some organizational and clarity issues also affect the presentation. Nonetheless, the paper addresses an important and under explored area, making it a worthwhile contribution that justifies acceptance